# A Forget-and-Grow Strategy for Deep Reinforcement Learning Scaling in Continuous Control

**Zilin Kang** [* 1 2] **Chenyuan Hu** [* 3] **Yu Luo** [4] **Zhecheng Yuan** [3 1] **Ruijie Zheng** [5] **Huazhe Xu** [3 1 6]

## Abstract

Deep reinforcement learning for continuous control has recently achieved impressive progress. However, existing methods often suffer from primacy bias—a tendency to overfit early experiences stored in the replay buffer—which limits an RL agent's sample efficiency and generalizability. In contrast, humans are less susceptible to such bias, partly due to *infantile amnesia*, where the formation of new neurons disrupts early memory traces, leading to the forgetting of initial experiences (Akers et al., 2014). Inspired by this dual processes of forgetting and growing in neuroscience, in this paper, we propose *Forget and Grow* (**FoG**), a new deep RL algorithm with two mechanisms introduced. First, *Experience Replay Decay (ER Decay)*—"forgetting early experience"—which balances memory by gradually reducing the influence of early experiences. Second, *Network Expansion*—"growing neural capacity"—which enhances agents' capability to exploit the patterns of existing data by dynamically adding new parameters during training. Empirical results on four major continuous control benchmarks with more than 40 tasks demonstrate the superior performance of FoG against SoTA existing deep RL algorithms, including BRO, SimBa and TD-MPC2.

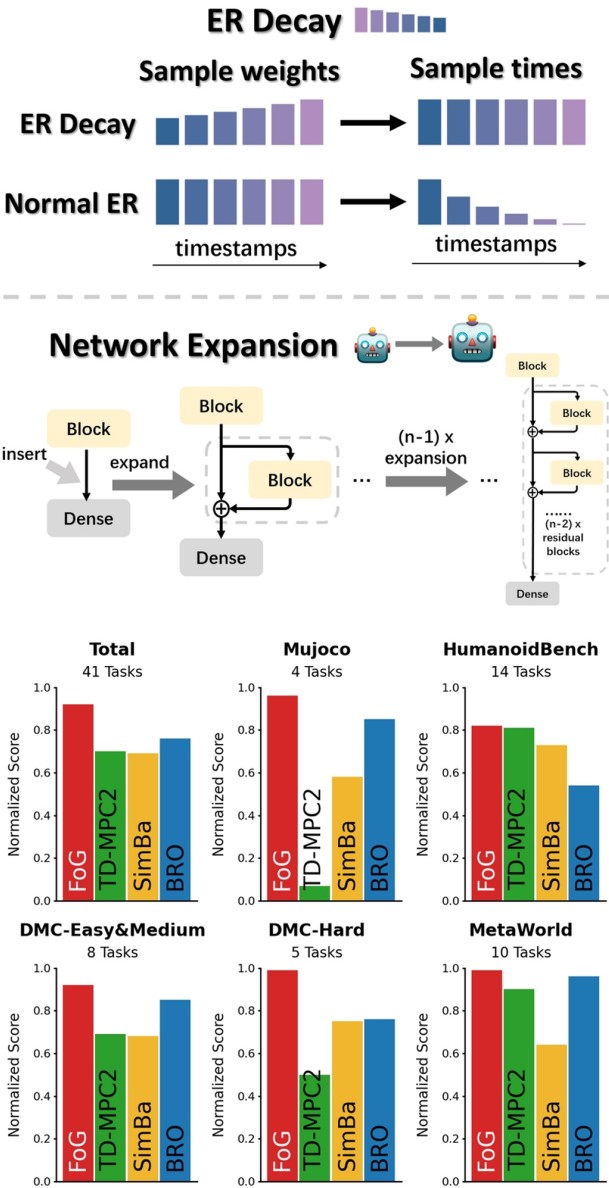

*Figure 1.* **Overview. Top:** we illustrate two key components of our strategy: ER Decay and Network Expansion. **Bottom:** comparison of normalized score. FoG outperforms popular model-based and model-free methods including TD-MPC2, SimBa and BRO.

---
[*]Equal contribution [1]Shanghai Qi Zhi Institute [2]Department of Computer Science and Technology, Tsinghua University [3]Institute for Interdisciplinary Information Sciences, Tsinghua University [4]Huawei Noah's Ark Lab [5]Computer Science, University of Maryland [6]Shanghai Artificial Intelligence Laboratory. Correspondence to: Zilin Kang <kzl22@mails.tsinghua.edu.cn>, Chenyuan Hu <cy-hu22@mails.tsinghua.edu.cn>, Huazhe Xu <huazhe_xu@mail.tsinghua.edu.cn>.

*Proceedings of the $42^{nd}$ International Conference on Machine Learning*, Vancouver, Canada. PMLR 267, 2025. Copyright 2025 by the author(s).

# 1. Introduction

Do humans remember learning how to speak or walk? For the majority, the answer is no. This phenomenon, known as **infantile amnesia** in neural science (Josselyn & Frankland, 2012), occurs because the hippocampus generates a large number of new neurons during infancy, which disrupts existing memory traces and leads to forgetting (Alberini & Travaglia, 2017). Observed in humans and other mammals, this phenomenon plays a critical role in the development of memory and learning abilities.

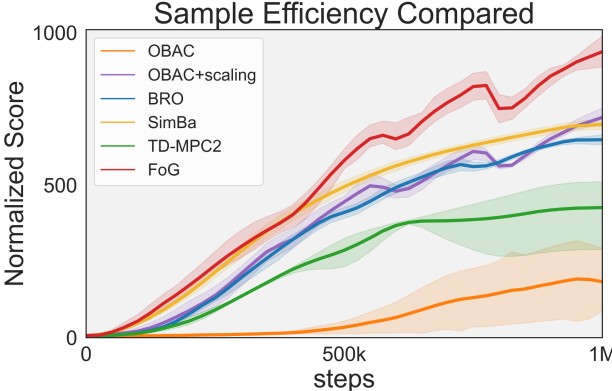

*Figure 2.* Normalized scores of algorithms on **DMC-Hard tasks (5 hardest Dog & Humanoid tasks)**. The performance of the OBAC+scaling method is comparable to that of SimBa and BRO, but when combined with FoG, it achieves superior results.

However, in the field of deep reinforcement learning, agents typically do not have a natural mechanism to forget their early training experiences. Instead, they often overfit to initial data, which is often described as primacy bias (Nikishin et al., 2022b; Qiao et al., 2023). In particular, deep RL methods that rely heavily on experience replay (Mnih et al., 2015; Fedus et al., 2020) with high replay ratios (D'Oro et al., 2022) tend to repeatedly revisit old transitions, reinforcing patterns formed in the early stage of training.

To address this issue, previous works (Nikishin et al., 2022b; Qiao et al., 2023) have introduced a reset mechanism that periodically resets part of the policy's network parameters. While resets partially reduce primacy bias, the older samples are still replayed more frequently than the newer ones. Thus, this imbalance could still lead to overfitting on the old experience, harming the overall performance.

To draw a parallel, infantile amnesia involves two key aspects—*forget* and *grow*. During infancy, the brain generates a large number of new neurons, which not only disrupt existing memory traces and lead to forgetting but also provide the capacity for reorganizing and forming new structures critical for memory and learning. This phenomenon inspires

our question: *can reinforcement learning agents follow a similar process, combining forgetting and growth, to mitigate primacy bias and improve performance?*

Our answer is *affirmative*, with two novel methods: **Experience Replay Decay (ER Decay)** and **Network Expansion**, both of which are simple and efficient. **ER Decay** reduces the sampling probability of older data in the replay buffer, effectively allowing the agent to "forget" outdated transitions in a way analogous to the memory disruption observed in infantile amnesia. Meanwhile, **Network Expansion** introduces new neurons to the model early in training, providing fresh capacity to adapt and reorganize, much like the growth of new neurons in infancy.

Together, these methods are directly inspired by the "forget and grow" mechanism observed in infantile amnesia, and they work in tandem to suppress primacy bias and enhance overall performance.

In this work, we give theory-based intuition on how does such a "forgetting plus growth" mechanism work, and we provide a thorough empirical investigation of its effectiveness. We also propose a new algorithm Forget-and-Grow (**FoG**), which integrates these methods into the OBAC algorithm (Luo et al., 2024) and further boosts performance using scaled networks and replay ratios. Our approach achieved highly competitive results across more than 40 environments in several benchmarks, including Mujoco (Todorov et al., 2012), DMControl (Tassa et al., 2018), Meta-World (Yu et al., 2020), and HumanoidBench (Sferrazza et al., 2024) surpassing popular methods including SimBa (Lee et al., 2024), BRO (Hansen et al., 2024) and TD-MPC2 (Hansen et al., 2024) in multiple settings.

To summarize, the contributions of this paper are three-fold:

1. We show empirically that reset mechanisms alone cannot fully resolve the primacy bias issue.

2. We introduce two strategies: ER Decay and Network Expansion, demonstrating their effectiveness in mitigating primacy bias.

3. Develop a new deep continuous control algorithm **FoG**, achieving state-of-the-art performance across several benchmarks.

# 2. Related Works

**Off-policy RL.** Off-policy reinforcement learning is a frequently used paradigm where agents learn policies from data generated by previous policies (Mnih et al., 2015; Munos et al., 2016; Prudencio et al., 2023; Ma et al., 2024), allowing for more efficient use of prior experiences. Due to the advantage of improving sample efficiency, it is widely used in scenarios where collecting on-policy data is costly

or risky. Many approaches focus on real-world application (Delarue et al., 2020; Yang et al., 2022) and algorithmic improvements such as reducing bias in Q-value estimation (Fujimoto et al., 2018; Lan et al., 2021), better utilizing offline datasets (Fujimoto et al., 2019; Schaul et al., 2016), and integration with other paradigms (Luo et al., 2024; Tan et al., 2024).

**Primacy bias.** The concept of *primacy bias* in deep reinforcement learning (RL) refers to the overfitting of policies to earlier experiences when training on progressively growing datasets, which can negatively impact the following learning process (Nikishin et al., 2022a). This phenomenon is particularly problematic under high replay ratios, where policies overfit to out-of-distribution data from past experiences, as noted by Li et al. (2023); Lyu et al. (2023). One straightforward approach to mitigate primacy bias is re-initializing the network to restore plasticity, as explored by Nikishin et al. (2022b); Ma et al. (2023); Nauman et al. (2024b). There are other methods to alleviate the problem including model ensembles (Chen et al., 2021), regularization (Kumar et al., 2023b), *plasticity injection* (Nikishin et al., 2023), and *ReDo* (Sokar et al., 2023). These approaches aim to balance stability and adaptability, reducing the impact of primacy bias and enhancing overall learning performance

**Experience replay.** To better utilize the previous experience and improve sample efficiency, Lin (1992) propose the concept of experience replay, which revisits transitions in the replay buffer with a uniform sampling strategy to update the agent. Following Lin (1992), Prioritized Experience Replay (Schaul et al., 2016) measures the priority of transitions according to the magnitude of their temporal-difference (TD) error so that the agent can focus on transitions that are more important to improve sample efficiency. Andrychowicz et al. (2018) introduces Hindsight Experience Replay (HER) which incorporates a set of additional goals into each trajectory to avoid complicated reward engineering. Zhang & Sutton (2018) proposes Combined Experience Replay (CER) that adds the latest transition to the batch and uses the corrected batch to train the agent. Corrected Uniform Experience Replay (CUER) (Yenicesu et al., 2024) also adopts the idea of balancing the sampling of the transitions in the replay buffer to make the sampling distribution more uniform considering the whole training process.

**Model capacity improvement in RL.** The most straightforward way to improve model capacity is model size scaling (Hestness et al., 2017). However, in RL, naive scaling can lead to instability or degraded performance (van Hasselt et al., 2018; Sinha et al., 2020; Bjorck et al., 2022). High-capacity models have shown effectiveness in offline RL (Kumar et al., 2023a; Lee et al., 2022) and model-based RL (Hafner et al., 2024; Hansen et al., 2024; Hamrick

et al., 2021). As for off-policy RL, model size scaling has exhibited advantages for both discrete action representation (Schwarzer et al., 2023) and continuous control (Nauman et al., 2024b).

Besides scaling, internal structural changes, such as activation functions and normalization, also improve capacity. For example, TD-MPC2 (Hansen et al., 2024) enhanced model performance by incorporating LayerNorm to stabilize gradients, Mish as a smoother activation function, and SimNorm to maintain stable updates across layers. Similarly, Nauman et al. (2024a) demonstrated that LayerNorm and residual connections significantly enhance performance, while SimBa (Lee et al., 2024) used running statistics normalization, residual feedforward blocks, and post-Layer normalization to address simplicity bias. These modifications highlight that structural improvements, alongside careful scaling, are crucial for leveraging high-capacity models effectively in RL.

## 3. Method

### 3.1. A Motivating Example

Primacy bias has been studied in previous works (Nikishin et al., 2022b; Qiao et al., 2023), and a common approach to mitigate it is to adopt a reset strategy. However, even if an agent resets multiple times during training, experience replay remains imbalanced: older transitions dominate the sampling process. The following theorem formalizes this issue:

**Theorem 3.1.** *Given a uniformly sampled replay buffer $\mathcal{D}$ that stores $N$ sequentially added transitions $\{\kappa_1, \kappa_2, \cdots, \kappa_N\}$, the earliest transition is sampled $\Omega(\beta \log N)$ times in expectation, where $\beta$ (the product of replay ratio and batch size) is a constant.*

*More specifically, for any transition $\kappa_t$ (with $t > 1$), its expected number of samples $\mathbb{E}[n_t]$ satisfies:*

$$\ln \frac{N}{t-1} + \frac{1}{N} - 1 < \frac{\mathbb{E}[n_t]}{\beta} < \ln \frac{N}{t-1} + 1 - \frac{1}{t-1}.$$

*Proof.* See Appendix A. □

This result indicates that older transitions are sampled considerably more often, regardless of how frequently resets happen. In fact, shorter reset intervals often lead to higher replay ratios and can even exacerbate primacy bias. To verify this, we train four agents on *humanoid-walk* and *HalfCheetah-v4* with a replay ratio of 10 and batch size of 256 (i.e., $\beta = 2560$). To ensure stability of training under such a high replay ratio, we add a layernorm after every dense layer.

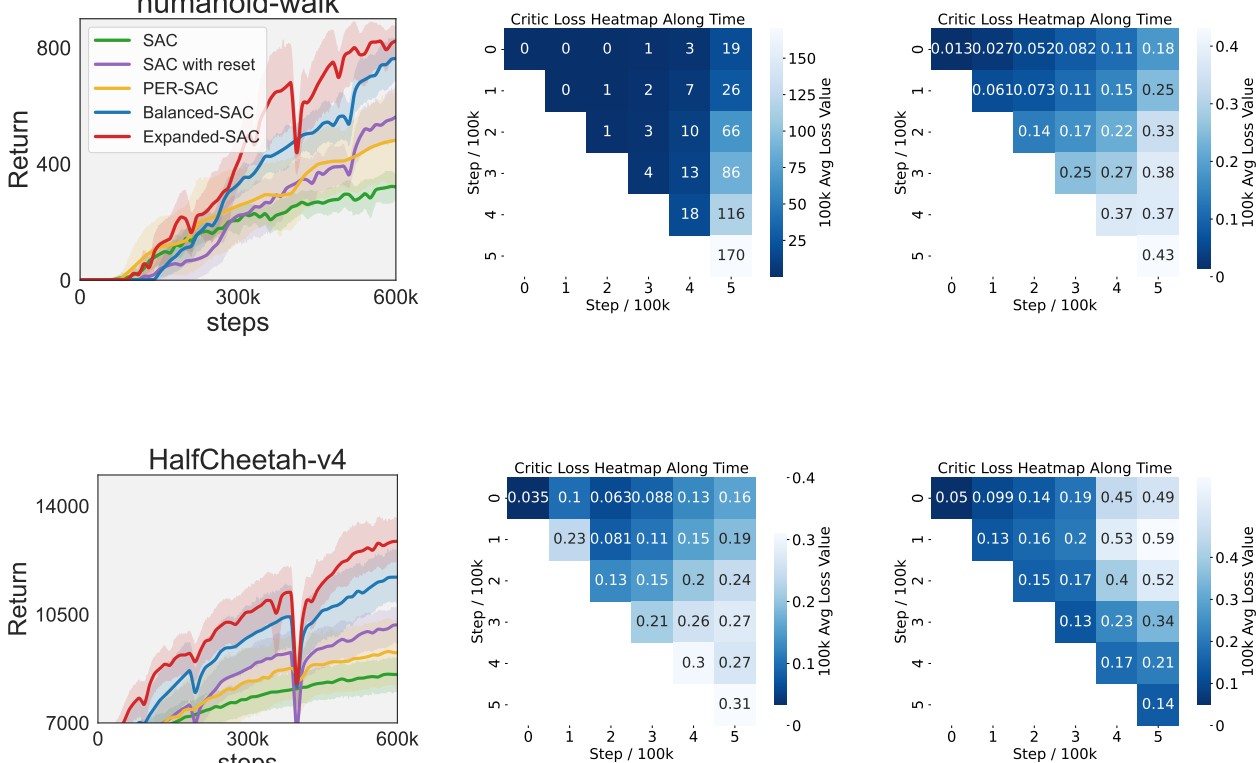

*Figure 3.* **Learning Curves and Heat Maps of SAC Variants. Top Left:** Return curves of various SAC variants in *humanoid-walk*. **Top Center:** Critic loss heatmap of Normal SAC in *humanoid-walk*. **Top Right:** Critic loss heatmap of SAC with reset in the humanoid-walk environment. **Bottom Left:** Return curves of various SAC variants in the *HalfCheetah-v4*. **Bottom Center:** Critic loss heatmap of Balanced SAC in *humanoid-walk*. **Bottom Right:** Critic loss heatmap of Expanded SAC in *humanoid-walk*. **About Critic Loss Heatmaps:** Every 100k steps, we measure critic loss over the entire buffer and average the loss every 100k steps to get critic loss heatmaps. The darker the color near the diagonal, the less influenced by primacy bias; conversely, the darker the color towards the top-right corner indicates greater influence from primacy bias.

- **SAC**: A baseline Soft Actor-Critic agent (Mnih et al., 2015) with uniform replay buffer and no resets.

- **SAC with reset**: SAC that resets at 15k, 50k, 100k, 200k, 400k, 600k, and 800k steps.

- **PER-SAC**: Based on SAC with reset, plus PER (prioritized experience replay) (Schaul et al., 2016).

- **Balanced SAC**: Based on SAC with reset, plus our ER decay mechanism to mitigate primacy bias, with $\epsilon = 1e-4, \tau = 1e-1$.

- **Expanded SAC**: Based on SAC with reset, plus both ER decay and network expansion, which expands critic networks from 3 dense layers to 7 dense layers, at 50k and 200k network iterations after each reset, 2 layers at a time.

We measure the critic loss across the buffer every 100k steps to generate critic loss heatmaps. As shown in Figure 3, the critic loss of **Normal SAC** is significantly higher than that of other SAC variants that incorporate resets, indicating its failure to fit the data after millions of updates. Even **SAC with reset** experiences a spike in loss in the diagonal areas as training progresses. This suggests that, despite multiple resets, the **SAC with reset** agent still overfits to early data and fails to adapt effectively to newer transitions. This observation highlights that *resetting alone is insufficient to mitigate primacy bias.*

In contrast, **Expanded SAC** exhibits a darker diagonal area, indicating that the model is less influenced by primacy bias, with little to no increase in loss over time. Not only does **Expanded SAC** reduce the loss on recent transitions more effectively than **SAC with reset**, but it also achieves superior

performance across both tasks. Specifically, it delivers **53%** and **27%** improvements in final scores on *humanoid-walk* and *HalfCheetah-v4*, respectively, compared to **SAC with reset**. Furthermore, **Expanded SAC** outperforms recent methods like *SimBa* (Lee et al., 2024) by **74%** and **22%**, while maintaining a simpler design.

Although this small-scale experiment is not exhaustive, it underscores the critical role of addressing primacy bias in experience replay and provides strong motivation for our proposed method.

### 3.2. Experience Replay Decay and Network Expansion

**Experience replay decay.** We incorporate a decay factor into experience replay so that the sampling probability of older transitions gradually decreases. This strategy partially "forgets" older samples and mitigates primacy bias.

**Theorem 3.2.** *Let $\mathcal{D}$ be a replay buffer with ER decay $\epsilon$. For any transition $\kappa_i$ in $\mathcal{D}$, the expected number of times it is sampled, $\mathbb{E}[n_i]$, is bounded by a constant $C$.*

*Proof.* See Appendix A. □

This result implies that the sampling frequency of older transitions stays within a constant range. However, a purely exponential decay quickly diminishes the sample weight of older transitions, causing them to virtually disappear from the replay buffer. This effectively reduces the buffer size and, in practice, can harm the final performance. Therefore, we set a lower bound for sampling weights:

$$w_{\{t,i\}} = \max\big(\tau, (1-\epsilon)^{t-i}\big),$$

where $\epsilon$ is the decay rate, $\tau$ is the minimum weight, and the sampling probability of transition $\kappa_i$ at time $t$ is:

$$P_{\{t,i\}} = \frac{w_{\{t,i\}}}{\sum_{j=1}^{t} w_{\{t,j\}}}.$$

Simulation (see Figure 4) shows that ER decay can effectively suppress sample times of older transitions and balance the sample times of transitions across a large range of steps.

During the experiments, we compared the performance of our ER decay method with Prioritized Experience Replay (PER). PER assigns higher replay weights to transitions with larger TD errors. In the previously discussed motivating example, we analyzed the loss landscape of both PER and ER decay. As demonstrated in Figure 5, the value in the diagonal row of ER Decay is generally smaller than those of PER, which indicates that ER Decay may better alleviate primacy bias. Experimentally, ER decay achieved a significant advantage over PER in the motivating example. This result was further validated through ablation studies conducted on a broader range of scenarios.

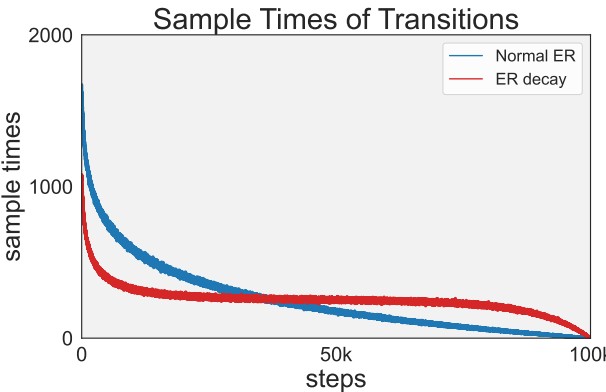

*Figure 4.* Sample times of transitions in a normal buffer and a decayed buffer with $\epsilon = 1e - 4, \tau = 0.01$ over 100k steps.

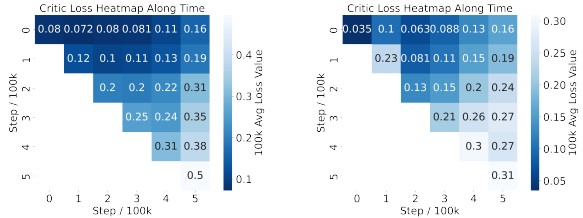

*Figure 5.* Changes in critic loss over time for PER(**left**) and ER decay(**right**) in *humanoid-walk*. The darker the color near the diagonal, the less influenced by primacy bias; conversely, the darker the color towards the top-right corner indicates greater influence from primacy bias.

**Network expansion.** Early training relies heavily on initial transitions in replay buffer, which often deviate significantly from the final policy distribution. Yet during this stage, the neural network has the highest plasticity. To address this, we propose *network expansion*, inspired by *infantile amnesia* in mammals. The method involves gradually adding new parameters to the critic network early in training (e.g., after each reset) through residual connections. These newly added parameters are not influenced by early transitions, enabling better adaptation to data shifts when combined with ER decay.

Our network structure is based on BRO's design, which incorporates layer normalization after each dense layer and residual connections for parameter management. Building on this, we modularized the network into distinct blocks, facilitating the implementation of network expansion.

Although starting with a large network can provide better

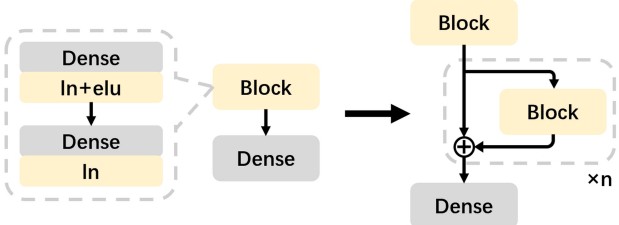

*Figure 6.* Network expansion illustration. We initialize networks with fewer parameters and progressively add a new block (in the frame) to residual connections at each expansion step.

performance in the early stages of training, it tends to encounter early convergence issues in the mid-to-late stages. In contrast, **agents trained with network expansion adapt more effectively to changing training objectives**. This observation is further validated in the FoG algorithm. Even when training begins with a network significantly larger than the one used in the motivating example, without network expansion, the agent is highly prone to loss explosions under larger replay ratios. This directly impacts the agent's stability. Therefore, network expansion is a necessary component in FoG. For more details, refer to Section 4.

A similar idea, called *plasticity injection* (Nikishin et al., 2023), also involves network expansion. However, compared to the complex parameter adjustments required by plasticity injection, our approach is simpler to implement, imposing minimal constraints. Our experiments demonstrate its superior performance and efficiency.

### 3.3. The Forget and Grow Strategy (FoG)

We combine **ER decay** and **network expansion** into a unified algorithm, FoG, built on the following key ideas:

**OBAC backbone.** Our FoG is based on the Offline Boosted Actor-Critic (Luo et al., 2024). OBAC enhances online policies using offline data and demonstrates strong performance across various standard benchmarks.

**Scaled critic networks and replay ratio.** For scaling up the networks, we utilize a larger network for the critic. To increase flexibility, we modularized the network structure, constructing the critic entirely from blocks connected via residual connections. Each block contains two dense layers, each followed by a layer normalization. This modular architecture proves highly compatible with our FoG mechanism.

Additionally, we increase the replay ratio to 10 and introduce a reset list to manage the agent's reset behavior effectively.

**ER decay and network expansion.** We introduce ER decay into the replay buffer, where older transitions are assigned lower sampling probabilities, and apply network expansion to the critic networks early in training. Together, these methods embody the concept of forget and grow, allowing the agent to better adapt to shifts in replay data. While simply scaling up OBAC yields performance on par with SimBa or BRO, the forget-and-grow approach is crucial for FoG's superior performance.

## 4. Experiment

To evaluate the performance of **FoG**, we collect in total **41** tasks from **4** domains: Mujoco (Todorov et al., 2012), DM-Control (Tassa et al., 2018), Meta-World (Yu et al., 2020), and HumanoidBench (Sferrazza et al., 2024), covering a wide range of challenges, including high-dimensional states and actions, sparse rewards, multi-object and delicate manipulation, and complex locomotion. The implementation details and environment settings are provided in Appendix B.

**Baselines.** We compare FoG against 3 state-of-the-art off-policy RL algorithms, including 2 model-free methods and 1 model-based method. Our baselines contain: 1) BRO (Nauman et al., 2024b), which scales the critic network of SAC while integrating distributional Q-learning, optimistic exploration, and periodic resets. 2) SimBa (Lee et al., 2024), which adopts running statistics normalization, residual feed-forward blocks, and post-layer normalization to address simplicity bias. 3) TD-MPC2 (Hansen et al., 2024), a high-efficient model-based RL method that combines model predictive control and TD-learning.

### 4.1. Experimental Results

Figure 7 presents the learning curves that demonstrate the performance of FoG alongside various baselines across diverse task suites. Overall, we observe that FoG typically outperforms most model-free and model-based baselines across various environments in terms of exploration efficiency and asymptotic performance. In HumanoidBench, FoG also exhibits comparable capabilities to the best baseline TD-MPC2.

Notably, with identical hyperparameters, FoG achieves consistently high performance across all benchmarks. Other baselines have certain weaknesses in some benchmarks. Due to the task-specific *done* signal setting, TD-MPC2 may perform poorly on Mujoco. SimBa gets lower scores in simple environments with small action dimensions, such as Mujoco and DMC-Easy while BRO performs worse in more complex environments with high action dimensions, such as DMC-Hard and HumanoidBench.

The key takeaway is that with very simple algorithmic

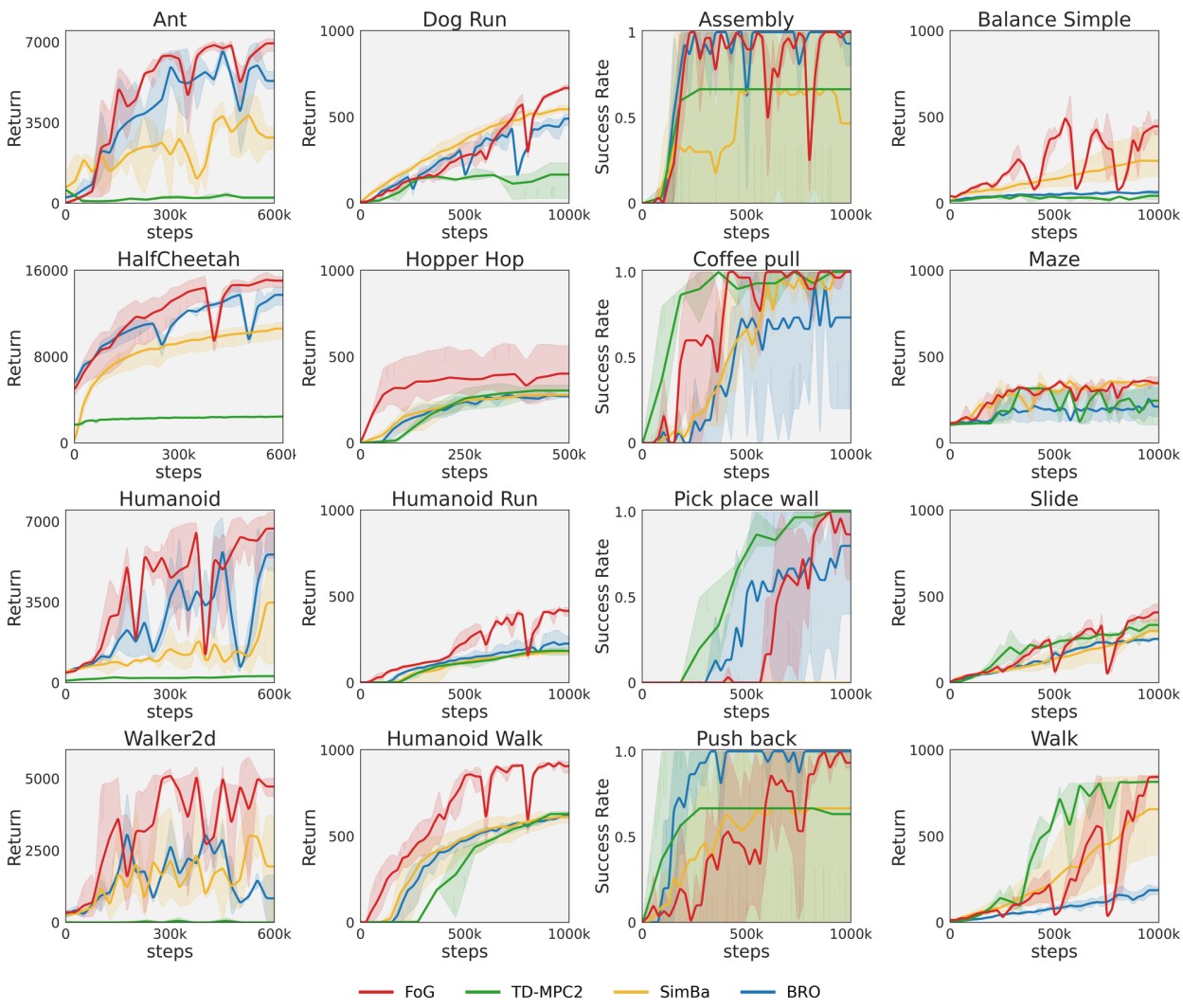

*Figure 7.* **Main results.** We provide performance comparisons for 16 of the 41 tasks, four for each task suite. Please refer to Appendix C for the comprehensive results. The solid lines are the average return/success rate, while the shades indicate 95% confidence intervals. All algorithms are evaluated with 3 random seeds.

changes, including ER decay, network expansion, and simple model structure modifications, FoG successfully overcomes primacy bias and achieves better performance. Notably, previous studies found that model performance saturated when the model parameters reached 5M (Nauman et al., 2024b). However, with the help of network expansion, FoG unlocks new levels of favorable model size scaling up to 23M parameters. We will discuss in detail the performance improvements from each modification in the upcoming ablation section.

### 4.2. Ablation Studies

We conduct several ablations to demonstrate the effectiveness of the design choices of FoG in this section.

**Choice of experience replay methods.** One of the key designs of our algorithm is ER decay, which gradually decreases the sampling weight of older transitions. To evaluate its effectiveness, we implement other experience replay methods including PER and CUER. For the sake of fairness, we simply replace the experience replay method of FoG while keeping other parts unchanged for comparison. Additionally, we also test the results without using any experience replay method. Results are shown in Figure 8. We observe that both using PER and ER decay significantly improve the final performance and convergence speed, with ER decay showing the best results. However, the improvement with CUER is marginal. In certain cases, CUER provides no improvement.

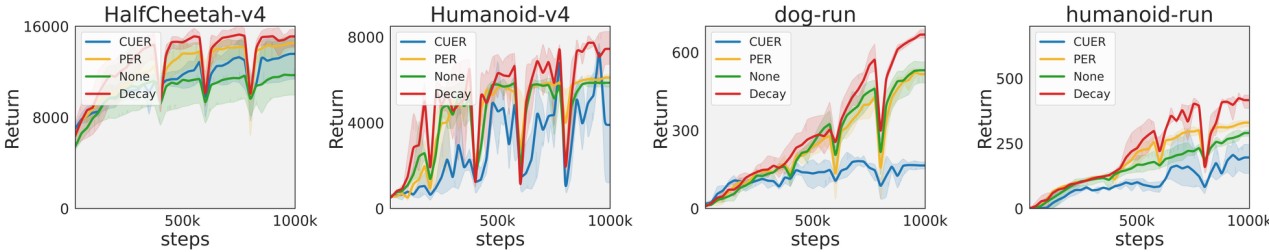

*Figure 8.* **Choice of experience replay methods.** We adopt 4 tasks from Mujoco and DMControl, two for each task suite, to compare different experience replay methods. Mean of 3 runs; shaded areas are 95% confidence intervals.

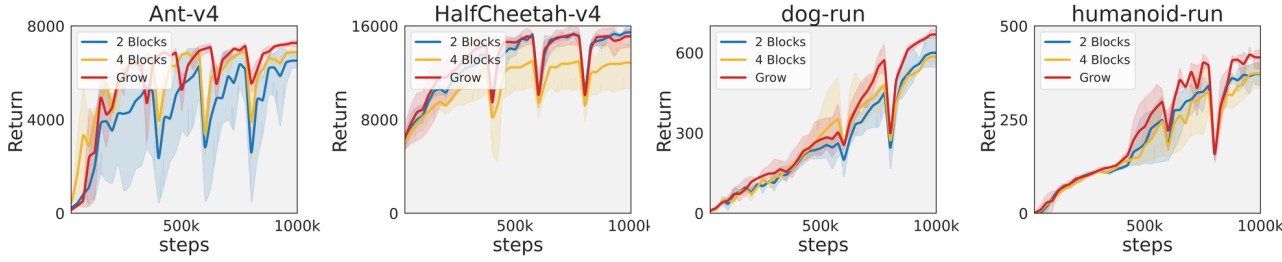

*Figure 9.* **Ablation on network expansion.** We adopt 4 tasks from Mujoco and DMControl, two for each task suite, to showcase the necessity of network expansion. Mean of 3 runs; shaded areas are 95% confidence intervals.

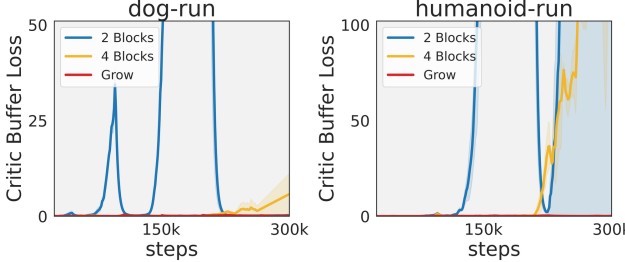

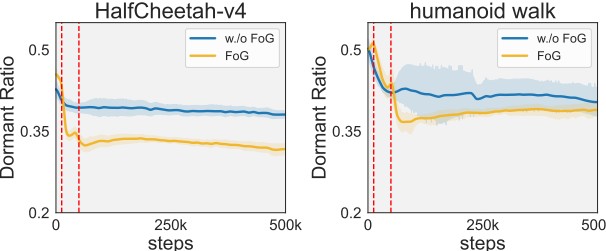

*Figure 10.* **Critic buffer loss with and without network expansion.** We visualize the critic buffer loss of the first 300k steps for the 2 tasks of DMControl suite. Mean of 3 runs; shaded areas are 95% confidence intervals.

*Figure 11.* **Dormant ratio during training.** We measure the ratio of dormant neurons during training on HalfCheetah-v4 and humanoid-walk. Red lines indicate time steps where network expansion happens.

**Necessity of network expansion.** To establish the importance of network expansion in our Framework of Growth (FoG), we conducted a series of experiments. Firstly, we compared the standard FoG, which incorporates dynamic network expansion, against versions with a fixed model size, denoted FoG *(fixed)*. These fixed models were configured with either 2 or 4 additional blocks, and network expansion was disabled. As illustrated in Figure 9, enabling network expansion leads to a discernible performance improvement over both fixed-size configurations. The critic buffer loss, presented in Figure 10, offers insight into this advantage: after resetting the network, the critic buffer network overfits early transitions and generate incorrect behavior cloning

signals, causing the loss to explode. However, network expansion helps to suppress this excessive catastrophic growth in the loss.

Furthermore, network expansion significantly enhances model plasticity by reducing the prevalence of dormant neurons. Activated neurons, characterized by non-zero gradients, can be updated by new data, whereas dormant neurons remain static during training. Consequently, a higher ratio of dormant neurons indicates a more severe loss of plasticity. This metric has been adopted in several recent studies as an indicator of a model's representational capacity and plasticity degradation (Liu et al., 2024; Sokar et al., 2023; Xu et al., 2023). Our experiments, shown in Figure 11, reveal that

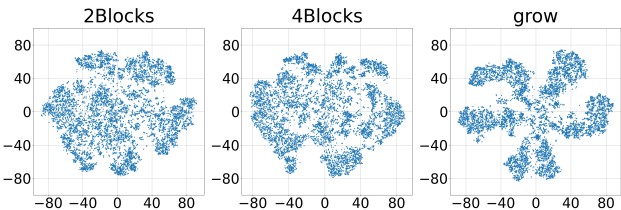

*Figure 12.* **T-SNE visualization of representations.** We visualize the representations via t-SNE after training 0.2M steps on HalfCheetah-v4. From left to right are the t-SNE results of 2 blocks, 4 blocks and expansion from 2 to 4 blocks.

FoG effectively reduces the number of dormant neurons. Notably, this reduction is achieved even when compared to baseline models that, despite possessing a larger overall parameter count than a fully expanded FoG agent, do not employ network expansion. This highlights the efficacy of the expansion mechanism itself. Moreover, steep drops in the dormant neuron ratio are consistently observed immediately following network expansion events, underscoring its direct role in reactivating parts of the network.

To further understand how the expanded network adapts to new memories and improves representations, we conducted a representation analysis on the HalfCheetah-v4. We sampled experiences from the replay buffer, passed them through the critic network, and extracted features from the final layer for t-SNE visualization in a 2D space. We compared three settings: FoG with network expansion, FoG (fixed) with 2 blocks, and FoG (fixed) with 4 blocks. As depicted in Figure 12, network expansion facilitates the formation of clearer clusters in the feature space. This observation suggests that network expansion enables the learning of better-separated and more structured representations, contributing to the overall performance gains.

## 5. Conclusion

In this work, we propose Forget-and-Grow (FoG), which effectively addresses the primacy bias problem in deep continuous control through two simple yet effective methods: ER decay and network expansion. By incorporating forget and grow, FoG enables agents to mitigate the overfitting to early experiences and boost their performance across various continuous control tasks. Abundant experiment results show the superiority of FoG compared with existing state-of-the-art off-policy RL and model-based RL algorithms, including BRO, SimBa and TD-MPC2. Our findings reveal a new perspective to alleviate the primacy bias, highlighting the potential of integrating inspired cognitive mechanisms into reinforcement learning frameworks. While FoG outperforms existing algorithms across various continuous control tasks, it is important to note that the increased computa-

tional complexity and longer training times may limit its practicality in scenarios that require rapid training. Additionally, the effectiveness of our two strategies is primarily supported by empirical experiments and intuitive theoretical insights, lacking a thorough and in-depth investigation into their mechanisms. Future works include seeking theoretical guarantees for the FoG strategies and applying the Forget-and-Grow strategy to a broader range of algorithms.

## Impact Statement

This work contributes to advancing the field of Reinforcement Learning (RL), particularly in the domain of off-policy RL algorithms. The proposed algorithm holds potential implications for real-world applications, especially in areas such as robotics. It's worth noting that exploration of an RL agent in real-world environments may require several safety considerations to avoid unsafe behavior during the process.

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

## A. Proofs

### A.1. Proof of Theorem 3.1

*Proof.* Given a uniformly sampled replay buffer $\mathcal{D}$ that stores $N$ sequentially added transitions $\{\kappa_1, \kappa_2, \cdots, \kappa_N\}$, transition $\kappa_t$ (with $t > 1$) is sampled with probability $\frac{1}{i}$, where $i > t$ is the number of transitions at sample time. So the expectation of sample times $\mathbb{E}[n_t]$ can be calculated as:

$$\mathbb{E}[n_t] = \beta(\sum_{i=t}^{N} \frac{1}{i}) = \beta(\sum_{i=1}^{N} \frac{1}{i} - \sum_{i=1}^{t-1} \frac{1}{i}) = \beta(H_N - H_{t-1})$$

where $H_N$ is the $N$-th harmonic number, and $H_{t-1}$ is the $(t-1)$-th harmonic number. And we have the following inequality:

$$\ln n + \frac{1}{n} < H_n < \ln n + 1$$

So we can get:

$$\ln \frac{N}{t-1} + \frac{1}{N} - 1 = (\ln N + \frac{1}{N}) - (\ln t - 1 + 1) < \frac{\mathbb{E}[n_t]}{\beta} < (\ln N + 1) - (\ln t - 1 + \frac{1}{t-1}) = \ln \frac{N}{t-1} + 1 - \frac{1}{t-1}$$

For the earliest transition $\kappa_1$, its expected sample times $\mathbb{E}[n_1]$ is bounded by:

$$\ln N + \frac{1}{N} < \frac{\mathbb{E}[n_1]}{\beta} < \ln N + 1$$

The earliest transition is at least sampled $\Omega(\beta \log N)$ times in expectation. □

### A.2. Proof of Theorem 3.2

*Proof.* Let $\mathcal{D}$ be a replay buffer with ER decay $\epsilon$. For any transition $\kappa_i$ in $\mathcal{D}$, its expected sample times $\mathbb{E}[n_i]$ can be calculated as:

$$\mathbb{E}[n_i] = \beta(\sum_{t=0}^{\infty} \frac{(1-\epsilon)^t}{1 + (1-\epsilon) + \cdots + (1-\epsilon)^{i+t-1}})$$

We have:

$$\mathbb{E}[n_i] \leq \mathbb{E}[n_1], \forall i \in \{1, 2, \cdots, N\}$$

So the expected sample times of any transition in $\mathcal{D}$ is bounded by the earliest transition.

$$\mathbb{E}[n_1] = \beta(\sum_{t=0}^{\infty} \frac{(1-\epsilon)^t}{1 + (1-\epsilon) + \cdots + (1-\epsilon)^t}) = \beta(\sum_{t=0}^{\infty} \frac{\epsilon(1-\epsilon)^t}{1 - (1-\epsilon)^{t+1}})$$

As $\epsilon = 1 - (1-\epsilon) < 1 - (1-\epsilon)^{t+1}$, we have:

$$\mathbb{E}[n_1] < \beta(\sum_{t=0}^{\infty} (1-\epsilon)^t) = \frac{\beta}{\epsilon}$$

So we can find a constant $C$ such that:

$$\mathbb{E}[n_i] < C, \forall i \in \{1, 2, \cdots, N\}$$

The expected sample times of any transition in $\mathcal{D}$ is bounded by a constant. □

## B. Implementation Details

### B.1. Hyperparameters

In this section, we delve into the specific implementation details of FoG. Our Hyperparameters are listed in Table 1.

Table 1. The hyperparameters of the proposed method

| Hyperparameters | Hyperparameter | Value |
|---|---|---|
| Hyperparameters | Optimizer (Critic) | AdamW |
| | Critic learning rate | 3e-4 |
| | Critic initial depth | 2 |
| | Critic maximal depth | 4 |
| | Critic expansion iters | 50k, 200k |
| | Optimizer (Actor) | Adam |
| | Actor dense layers | 3 |
| | Actor learning rate | 3e-4 |
| | Actor log std clipping | (-20,2) |
| | Discount factor | 0.99 |
| | Batch size | 256 |
| | Replay buffer size | 1e6 |
| | ER Decay $\epsilon$ | 1e-5 |
| | Minimal buffer weight $\tau$ | 0.1 |
| | Behavior clone weight $\lambda$ | 1e-3 |
| Network Architecture | Network hidden dim | 512 |
| | Network activation function | elu |
| | Critic depth | 2-4 |

For all tasks, we use a max-entropy framework (Haarnoja et al., 2018) for the online learning policy $\pi$ with automatic temperature tuning. Besides, we set the pessimism of the online policy (Moskovitz et al., 2022) to 0 in MetaWorld tasks to further encourage exploration. In other benchmarks it is set to 1.

We use two reset lists for FoG. For 4 relatively simple locomotion tasks (h1-stand, h1-walk, h1-stair and h1-slide) in HumanoidBench, we use a reset list of 15k, 50k, 250k, 500k, 750k (The same as BRO's reset list) to further improve exploitation. We use a reset list of 15k, 50k, 100k, 200k, 400k, 600k, 800k for other benchmarks and other tasks in HumanoidBench.

### B.2. Details of Network Expansion

The expansion of the critic network is a key component of FoG. Each expansion step adds a new block composed of two dense layers with 512 hidden dims and ELU activation functions. Surprisingly, **we find that there is no restriction on the initialization of the new block**, and we initialize the new block with the same initialization as the original network (which is orthogonal init with scale of $\sqrt{2}$).

At each expansion step, we reinitialize the optimizer and decay the learning rate by the number of parameters in the network. Namely, we use init_lr $\times \frac{\text{init\_params}}{\text{current\_params}}$ to decay the learning rate, where init_lr is the initial learning rate, init_params is approximated by the number of dense layers in the initial network, and current_params is the number of dense layers in the current network.

Each time the network is reset, we reinitialize the depth of the critic network back to 2, so the network can expand again.

### B.3. Other Implementation Details

**Small actor network** Previous work (Lee et al., 2024; Nauman et al., 2024b) has shown that scaling up the actor network only provides marginal improvements in performance. To simplify our framework, we did not impose any special characteristics or constraints on the actor network or optimizer beyond what is typically done in standard SAC implementations. Specifically, we used the simplest MLP architecture and the Adam optimizer, which also provides a solid foundation for deploying the actor.

**About OBAC implementation**    In the OBAC algorithm, an offline agent is used to improve the online agent. Specifically, the offline agent adds a constraint to the online actor, encouraging it to learn from the offline agent when the Q-value estimate of the online actor is lower than that of the offline agent. In our experiments, we found that after a reset during training, the offline agent could quickly gain an advantage over the online actor by better utilizing the information in the buffer. Allowing the actor to immediately learn from the offline agent could lead to early convergence. To address this, we introduced a "protection period" for the online actor, during which the OBAC algorithm is temporarily disabled after a reset. This period, which we call *OBAC wait*, was found to yield fine results when set to 250k network iterations in all cases. Thus, we use 250k as the default in all experiments.

### B.4. Is Forget and Grow a Universal Technique?

In the FoG algorithm, we use OBAC as the foundational framework, combined with scaling the network size and replay ratio. A natural question arises: is the Forget and Grow technique universally applicable?

We tested SAC in the MuJoCo environment and observed the following: even with the standard SAC algorithm at a replay ratio of 1, using ER decay "forget" technique led to stable performance improvements. However, network expansion required a larger replay ratio to achieve relatively significant effects. This may be due to the fact that the newly introduced parameters in network expansion require more intense updates before they become effective.

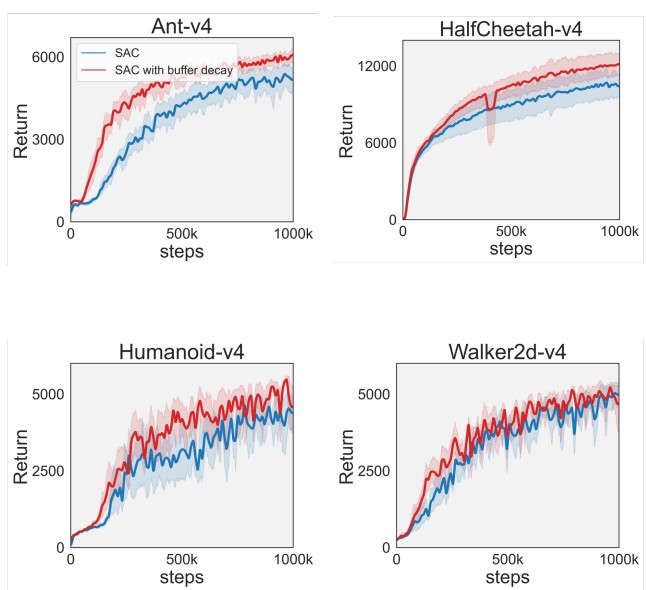

*Figure 13.* The results of SAC and SAC with ER decay **4** tasks in **Mujoco**.

We also evaluated the data efficiency of the FoG-enhanced SAC (FoG-SAC) algorithm in both DMC-Hard and MuJoCo environments. We found that FoG-SAC achieved data efficiency close to that of BRO, showing significant improvement in the later stages of training. However, it did not outperform BRO, which had undergone other algorithmic adjustments.

We also tested FoG with BRO, but FoG-BRO yielded performance very similar to that of the original BRO. This suggests that some of the adjustments in BRO may conflict with the FoG mechanism.

Overall, our experiments indicate that FoG integrates well with the native SAC and OBAC algorithms. However, its effectiveness when combined with other existing algorithms warrants further investigation.

### B.5. Baselines and Environments

We compare FoG with BRO, SimBa and TD-MPC2, we use their official implementations, hyperparameters and results to ensure a fair comparison.

*Figure 14.* The results of FoG-SAC

1. **BRO**: We use the official implementation from https://github.com/naumix/BiggerRegularizedOptimistic.

2. **SimBa**: We use SimBa-SAC from official implementation and results from https://github.com/SonyResearch/simba.

3. **TD-MPC2**: We use the official implementation and results from https://github.com/nicklashansen/tdmpc2.

4. **SAC**: We use implementation from https://github.com/proceduralia/high_replay_ratio_continuous_control.

We use the official setting of each task domain, including the reward setting, the task horizon, the done signal and there original state-action spaces.

### B.6. Official Implementation of FoG

Please check https://github.com/nothingbutbut/FoG.git for official implementation of FoG.

## C. Complete Experimental Results

To show the superiority of FoG, we list all the experimental results in this section.

### C.1. Numerical Results

*Table 2.* Normalized Score over benchmarks

| Environment | FoG | TD-MPC2 | SimBa | BRO |
|---|---|---|---|---|
| Mujoco | **0.96 ± 0.02** | 0.07 ± 0.00 | 0.58 ± 0.03 | 0.85 ± 0.04 |
| DMC-Easy | **0.99 ± 0.00** | 0.73 ± 0.01 | 0.74 ± 0.04 | 0.90 ± 0.11 |
| DMC-Medium | **0.85 ± 0.05** | 0.65 ± 0.03 | 0.63 ± 0.02 | 0.79 ± 0.04 |
| DMC-Hard | **0.99 ± 0.01** | 0.50 ± 0.02 | 0.75 ± 0.01 | 0.76 ± 0.02 |
| MetaWorld | **0.99 ± 0.00** | 0.90 ± 0.08 | 0.64 ± 0.07 | 0.96 ± 0.04 |
| HumanoidBench | **0.82 ± 0.02** | **0.81 ± 0.02** | 0.73 ± 0.04 | 0.54 ± 0.01 |
| Total | **0.92 ± 0.01** | 0.70 ± 0.02 | 0.69 ± 0.02 | 0.76 ± 0.02 |

*Table 3.* Returns of *600k* steps from Mujoco tasks

| Method | FoG | TD-MPC2 | SimBa | BRO |
|---|---|---|---|---|
| Ant | **6979.19 $\pm$ 98.06** | 563.04 $\pm$ 54.12 | 4827.24 $\pm$ 148.80 | 6798.11 $\pm$ 237.61 |
| HalfCheetah | **15119.91 $\pm$ 583.41** | 2445.42 $\pm$ 48.43 | 10393.10 $\pm$ 656.98 | 13765.24 $\pm$ 712.18 |
| Humanoid | **7170.79 $\pm$ 24.58** | 282.40 $\pm$ 25.21 | 2803.31 $\pm$ 290.75 | 6038.88 $\pm$ 918.07 |
| Walker2d | **5310.89 $\pm$ 320.60** | 93.38 $\pm$ 66.43 | 3365.35 $\pm$ 695.81 | 4223.97 $\pm$ 457.07 |

*Table 4.* Returns of *150k* steps from DMC-Easy tasks

| Method | FoG | TD-MPC2 | SimBa | BRO |
|---|---|---|---|---|
| Cartpole Balance | 999.62 $\pm$ 0.08 | 997.77 $\pm$ 0.58 | 999.64 $\pm$ 0.17 | **999.70 $\pm$ 0.22** |
| Cartpole Swingup | **879.58 $\pm$ 0.43** | 839.40 $\pm$ 35.20 | 872.01 $\pm$ 8.37 | 878.20 $\pm$ 0.88 |
| finger-spin | **983.07 $\pm$ 4.49** | 947.50 $\pm$ 27.89 | 703.57 $\pm$ 124.98 | 941.23 $\pm$ 20.54 |
| Hopper Stand | **922.97 $\pm$ 14.54** | 28.17 $\pm$ 28.21 | 237.10 $\pm$ 106.32 | 604.57 $\pm$ 416.54 |

*Table 5.* Returns of *500k* steps from DMC-Medium tasks

| Method | FoG | TD-MPC2 | SimBa | BRO |
|---|---|---|---|---|
| Acrobot Swingup | 423.88 $\pm$ 21.81 | 368.27 $\pm$ 39.63 | 377.23 $\pm$ 28.29 | **509.34 $\pm$ 38.23** |
| Hopper Hop | **409.42 $\pm$ 119.77** | 285.43 $\pm$ 56.43 | 278.30 $\pm$ 2.66 | 288.40 $\pm$ 9.97 |
| Humanoid Stand | **866.60 $\pm$ 27.14** | 401.87 $\pm$ 40.85 | 402.85 $\pm$ 46.63 | 769.53 $\pm$ 131.23 |
| Walker Run | **820.02 $\pm$ 2.45** | 818.07 $\pm$ 7.18 | 753.80 $\pm$ 10.10 | 760.53 $\pm$ 18.42 |

*Table 6.* Returns of *1000k* steps from DMC-Hard tasks

| Method | FoG | TD-MPC2 | SimBa | BRO |
|---|---|---|---|---|
| Dog Run | **652.78 $\pm$ 7.02** | 183.13 $\pm$ 14.75 | 534.02 $\pm$ 16.77 | 479.68 $\pm$ 12.26 |
| Dog Trot | **911.62 $\pm$ 7.98** | 423.13 $\pm$ 42.88 | 856.32 $\pm$ 26.84 | 842.09 $\pm$ 51.18 |
| Dog Walk | **954.09 $\pm$ 6.14** | 719.83 $\pm$ 55.70 | 924.67 $\pm$ 8.60 | 948.46 $\pm$ 4.95 |
| Humanoid Run | **436.34 $\pm$ 10.87** | 178.17 $\pm$ 2.93 | 177.10 $\pm$ 9.81 | 235.07 $\pm$ 48.55 |
| Humanoid Walk | **932.03 $\pm$ 6.96** | 572.10 $\pm$ 14.33 | 609.73 $\pm$ 36.53 | 609.56 $\pm$ 3.05 |

Table 7. Success rates of *1000k* steps from Metaworld tasks

| Method | FoG | TD-MPC2 | SimBa | BRO |
|---|---|---|---|---|
| Assembly | **1.00 ± 0.00** | 0.67 ± 0.47 | 0.70 ± 0.42 | **1.00 ± 0.00** |
| Coffee pull | **1.00 ± 0.00** | **1.00 ± 0.00** | **1.00 ± 0.00** | 0.93 ± 0.09 |
| Coffee push | 0.93 ± 0.05 | **1.00 ± 0.00** | 0.97 ± 0.05 | 0.87 ± 0.19 |
| Disassemble | **1.00 ± 0.00** | 0.67 ± 0.47 | **1.00 ± 0.00** | **1.00 ± 0.00** |
| Pick out of hole | **1.00 ± 0.00** | **1.00 ± 0.00** | **1.00 ± 0.00** | **1.00 ± 0.00** |
| Pick place | **1.00 ± 0.00** | **1.00 ± 0.00** | **1.00 ± 0.00** | **1.00 ± 0.00** |
| Pick place wall | **1.00 ± 0.00** | **1.00 ± 0.00** | 0.00 ± 0.00 | 0.80 ± 0.28 |
| Push back | **1.00 ± 0.00** | 0.67 ± 0.47 | 0.67 ± 0.47 | **1.00 ± 0.00** |
| Shelf place | **1.00 ± 0.00** | **1.00 ± 0.00** | 0.10 ± 0.14 | **1.00 ± 0.00** |
| Stick push | **1.00 ± 0.00** | **1.00 ± 0.00** | 0.00 ± 0.00 | **1.00 ± 0.00** |

Table 8. Returns of *1000k* steps from HumanoidBench tasks

| Method | FoG | TD-MPC2 | SimBa | BRO |
|---|---|---|---|---|
| Balance Hard | 72.84 ± 1.82 | 61.23 ± 2.83 | **79.87 ± 9.15** | 62.10 ± 2.48 |
| Balance Simple | **546.66 ± 66.01** | 52.88 ± 8.01 | 256.46 ± 135.38 | 68.77 ± 5.11 |
| Crawl | **971.02 ± 0.65** | 963.36 ± 2.48 | 939.58 ± 18.65 | 897.50 ± 31.20 |
| Hurdle | 86.31 ± 7.19 | **363.48 ± 34.69** | 208.65 ± 7.76 | 48.50 ± 0.65 |
| Maze | 380.13 ± 5.53 | 323.64 ± 4.12 | **389.39 ± 19.61** | 273.00 ± 61.73 |
| Pole | **817.70 ± 73.53** | 647.96 ± 172.64 | 754.56 ± 4.06 | 340.60 ± 24.14 |
| Reach | 3963.71 ± 289.43 | 3913.07 ± 740.94 | **4418.50 ± 395.17** | 3984.43 ± 236.95 |
| Run | 437.49 ± 49.97 | **780.80 ± 3.18** | 262.65 ± 73.32 | 49.33 ± 12.19 |
| Sit Hard | 814.95 ± 12.93 | 749.94 ± 14.30 | 667.38 ± 206.50 | **829.10 ± 4.34** |
| Sit Simple | 843.99 ± 8.91 | 801.01 ± 1.15 | **860.78 ± 5.00** | 850.03 ± 3.88 |
| Slide | **396.74 ± 25.44** | 329.21 ± 26.01 | 270.43 ± 17.06 | 250.10 ± 7.94 |
| Stair | 386.91 ± 117.06 | **562.50 ± 24.62** | 226.79 ± 162.50 | 77.57 ± 0.70 |
| Stand | 806.14 ± 27.59 | 812.80 ± 2.95 | **845.22 ± 9.60** | 799.73 ± 16.74 |
| Walk | **842.72 ± 8.53** | 813.88 ± 1.38 | 619.36 ± 200.68 | 186.17 ± 27.30 |

### C.2. Learning Curves

One thing to notice about DMC-easy tasks Figure 16 is that the starting point of the learning curve is 25k steps, which is because the first evaluation step of BRO is at 25k steps. At this time, FoGhas already achieved convergence in some environments, like *cartpole-swingup*.

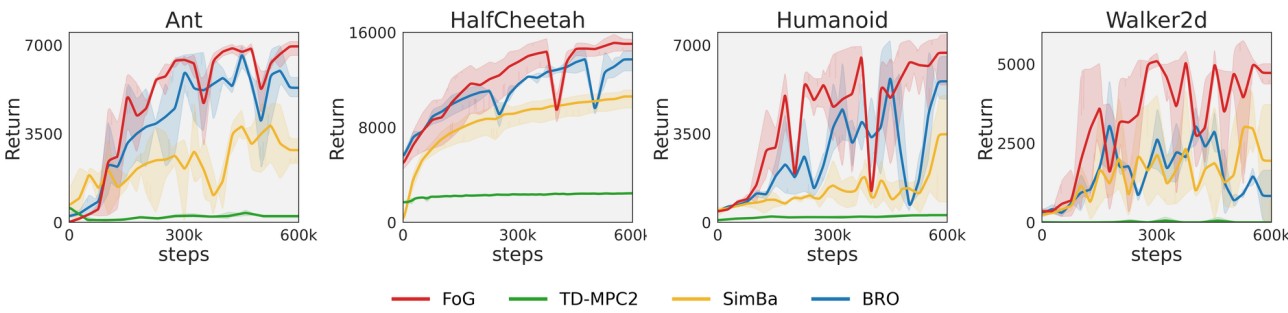

Figure 15. The results of **4** tasks in **Mujoco**.

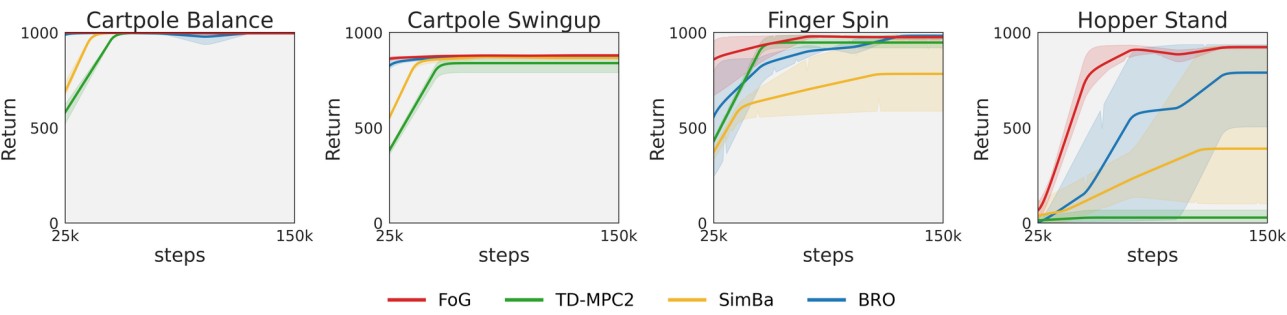

*Figure 16.* The results of **4** tasks in **DM Control Easy**.

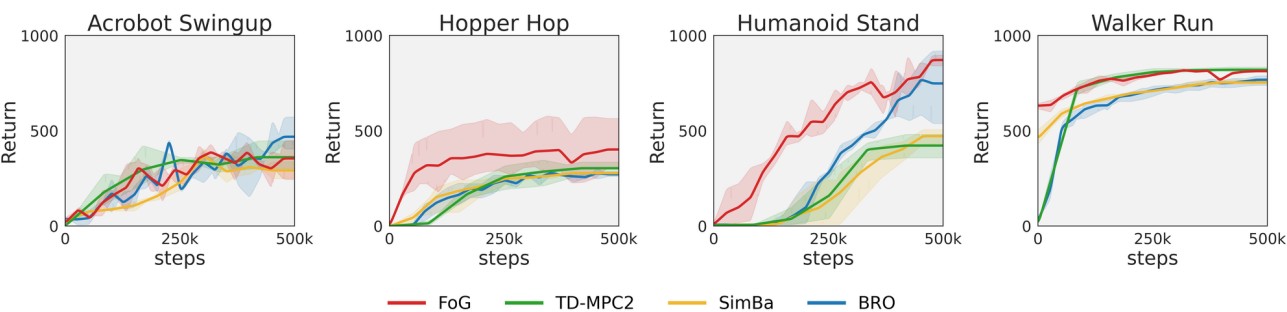

*Figure 17.* The results of **4** tasks in **DM Control Medium**.

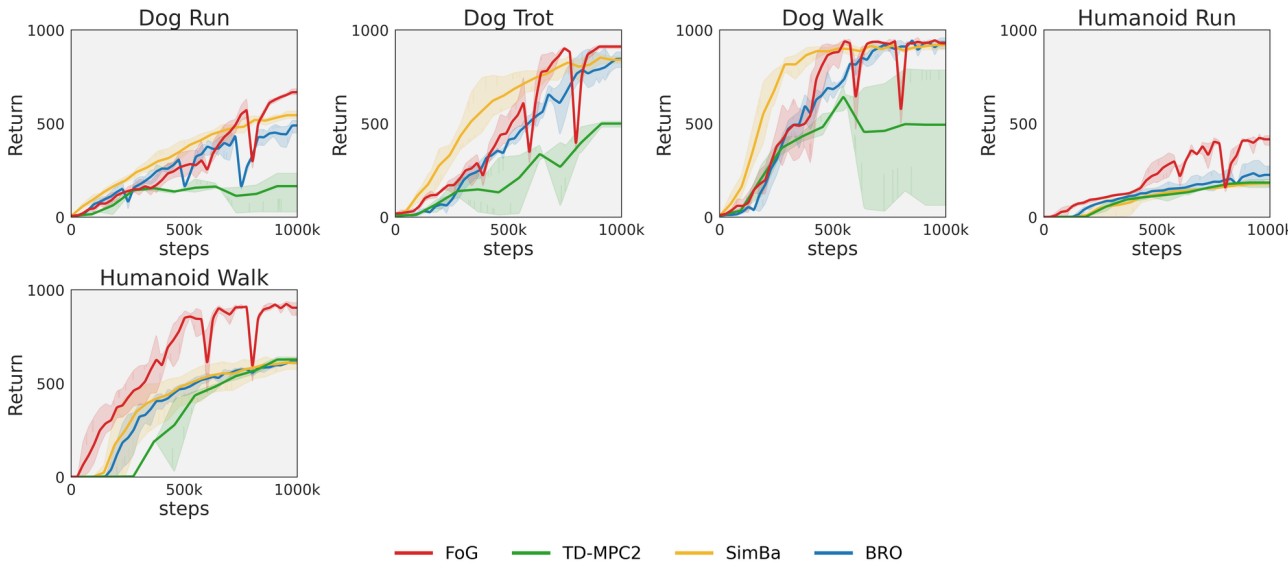

*Figure 18.* The results of **5** tasks in **DM Control Hard**.

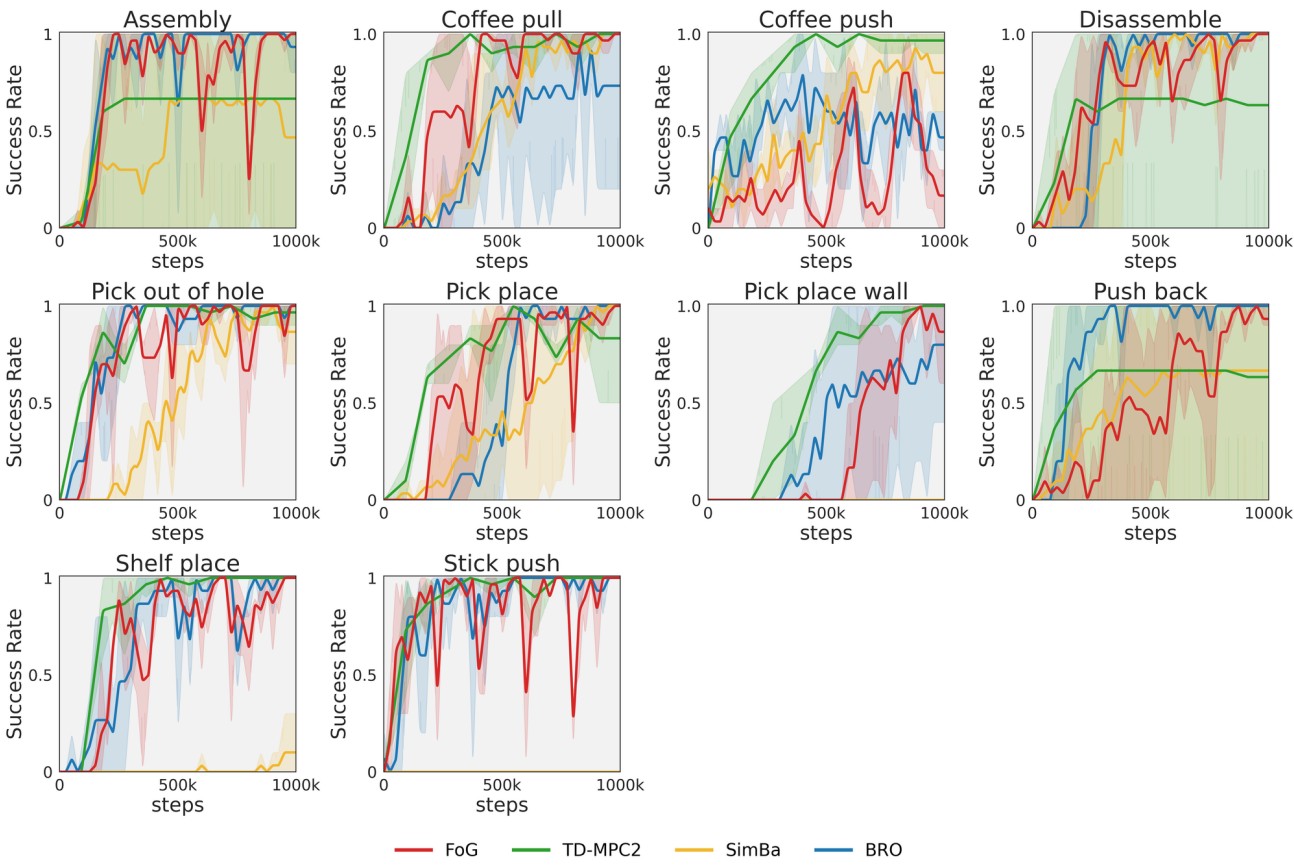

*Figure 19.* The results of **5** tasks in **Meta-World**.

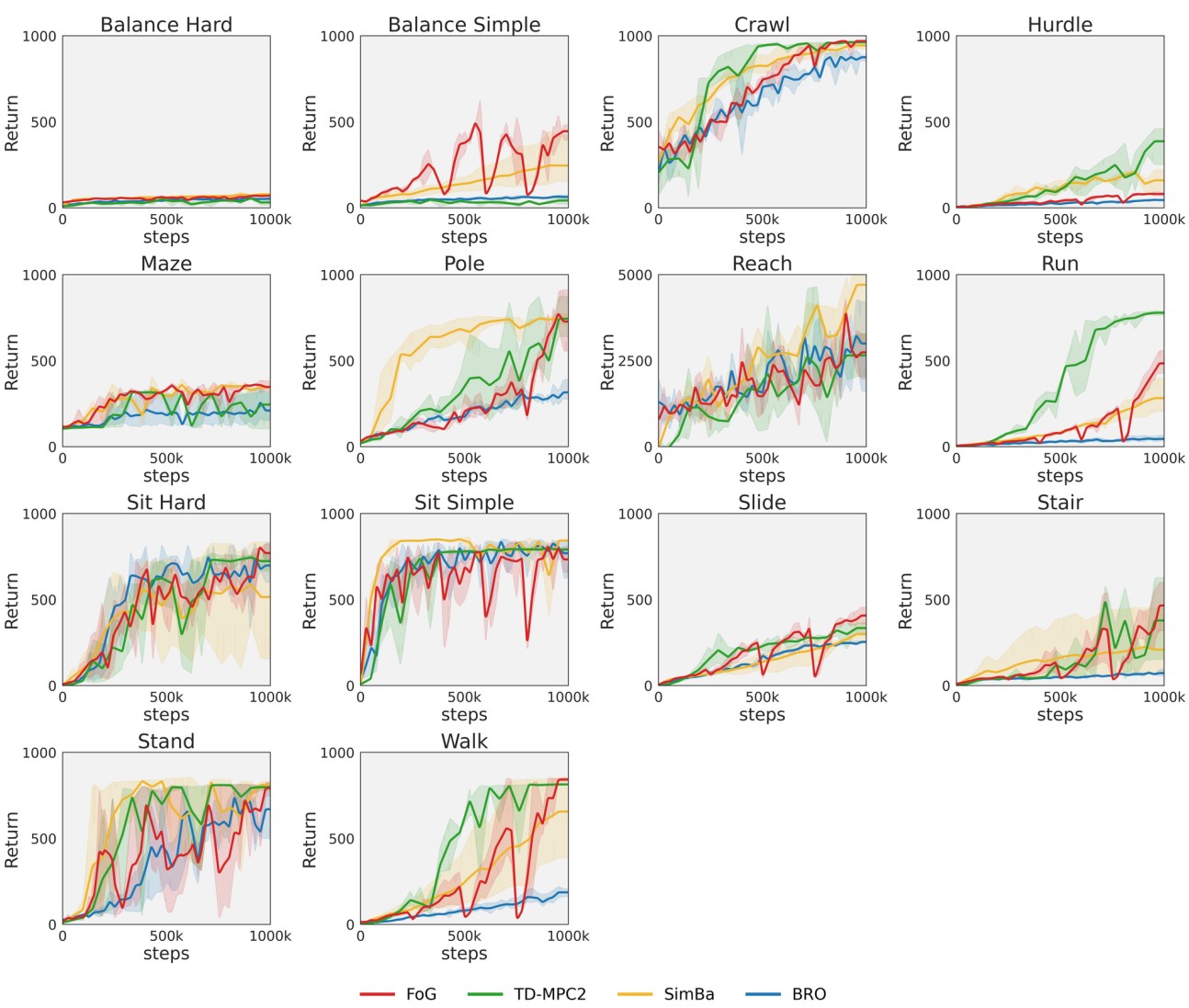

*Figure 20.* The results of **14** tasks in **HumanoidBench**.

## D. Performance comparison under similar computation cost

To demonstrate that the performance gains of FoG are not merely due to increased computation, we provide a performance comparison under similar computational costs.

We evaluate SimBa (depth=10, approximately 42M parameters), TD-MPC2 (19M version), and BRO (depth=10, approximately 42M parameters) on the humanoid benchmark to compare their performance against FoG, despite all of them having more parameters than FoG (at most 21M). Our experiments demonstrate that FoG outperforms competitive baselines even when compared to larger models, achieving superior performance while using less computation in this setup.

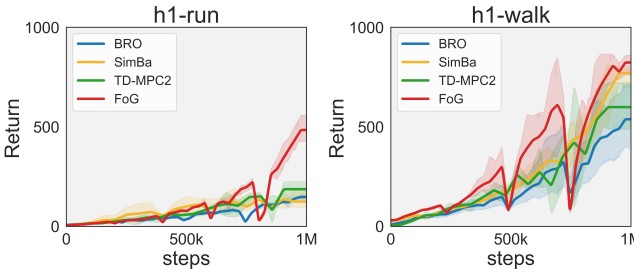

*Figure 21.* The results under similar computation cost

# E. Comparison to similar methods

We conducted a comparative analysis between FoG and the Neuroplastic Expansion (NE) algorithm(Liu et al., 2024), utilizing the official NE implementation to ensure fairness. The evaluation was performed across four diverse tasks: HalfCheetah-v4, dog-run, dog-walk, and humanoid-walk. Both the original NE model and a variant with expanded capacity were tested to examine the impact of model scaling on performance.

The results consistently demonstrate that FoG outperforms NE across all tasks, even when operating at relatively low update-to-data ratios. Notably, increasing the capacity of the NE model yields only marginal performance gains, whereas FoG maintains robust improvements without requiring significant model enlargement. These findings underscore FoG's superior scalability and adaptability, establishing it as a more effective approach for handling complex reinforcement learning environments.

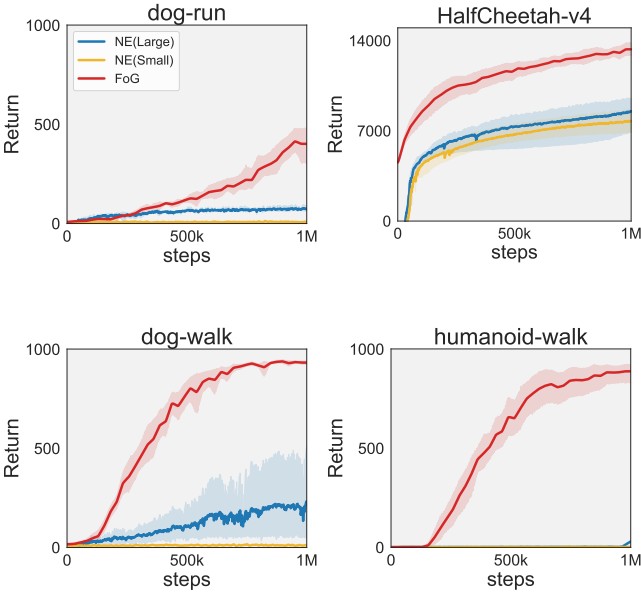

*Figure 22.* Comparison to Neuroplastic Expansion method

