# OpenReview forum: "A Forget-and-Grow Strategy for Deep Reinforcement Learning Scaling in Continuous Control"
_ICML.cc/2025/Conference — ICML 2025 poster_

### Official Review · Reviewer_6rRf · 2025-03-10

**Overall Recommendation:** 3

**Summary:**

Deep reinforcement learning suffers from primacy bias, a tendency to overfit early experiences stored in the replay buffer. This paper proposes Forget and Grow (FoG), a novel method with two new components: Experience Replay Decay (ER Decay) and Network Expansion. ER Decay gradually reduces the influence of early experiences, while Network Expansion dynamically adds new parameters during training. Comprehensive experiments on all kinds of continuous control tasks show that FoG outperforms the state-of-the-art methods.

**Claims And Evidence:**

Yes. The claims are supported by comprehensive experiments and ablation studies. The results are convincing.

**Essential References Not Discussed:**

No. The paper is well-positioned in the literature.

**Experimental Designs Or Analyses:**

Yes. The experiments are comprehensive and well-designed. The ablation study is also convincing.

**Methods And Evaluation Criteria:**

Yes.

**Other Comments Or Suggestions:**

There are large space in page 1 and 2, which can be reorganized to make the paper more compact.

**Other Strengths And Weaknesses:**

Strengths
- The paper is well-written and well-organized. The method is simple and effective. The experiments are comprehensive and convincing.
- Figures are beautiful and mostly informative.

Weaknesses
- The ER Decay in Figure 1 is not very clear. It looks like ER Decay will sample a transition more times than Normal ER, making me confused why it is called 'Decay'.

**Questions For Authors:**

- ER decay gradually decreases the sampling weight of older transitions, how to understand Figure 4 that ER decay is lower than Normal ER at the beginning and then higher than Normal ER? What does the x-axis 'steps' mean, environment step? Does the curve represent the sample times of a fix transition?
- Any intuition for the design of the Network Expansion? Are there any alternative ways to expand the network?
- line 37, Network Expansion introduces new neurons to the model early in training. Any implementation details about how to decide 'early in training'?
- What is the final size of the network after training compared to other methods?

**Relation To Broader Scientific Literature:**

The paper is related to the rl in continuous control tasks. The key contribution is to alleviate the primacy bias in deep reinforcement learning. The method is related to the experience replay and network expansion. The paper is well-positioned in the literature.

**Theoretical Claims:**

The proofs are reasonable. I did not check them in detail.

---

> ### Author Rebuttal · Authors · 2025-04-01
>
> We thank the reviewer for the time and effort you have dedicated to reviewing our work. We deeply appreciate your careful and thorough review. In the following, we seek to address each of your concerns.
> ___
> **Q1:** *"The ER Decay in Figure 1 is not very clear. It looks like ER Decay will sample a transition more times than Normal ER, making me confused why it is called 'Decay'."*
>
> **A:** The "sample times" in Figure 1 represent relative values, not absolute ones, so they should only be used for horizontal comparison. The term "Decay" actually refers to the fact that the probability of sampling transitions gradually decreases as training progresses. When a transition first enters the replay buffer, it is considered new and has a higher sampling probability. However, as training continues and the data ages, the probability of sampling that transition decreases over time.
> ___
> **Q2:** *"There are large space in page 1 and 2, which can be reorganized to make the paper more compact."*
> **A:** Thank you for the suggestion. We will make the necessary adjustments in the revised version to make the paper more compact.
> ___
> **Q3:** *"ER decay gradually decreases the sampling weight of older transitions, how to understand Figure 4 that ER decay is lower than Normal ER at the beginning and then higher than Normal ER? What does the x-axis 'steps' mean, environment step? Does the curve represent the sample times of a fix transition?"*
>
> **A:** The "steps" on the x-axis actually represent environment steps. The curve shows the total number of times that transitions collected at each environment step are sampled from the replay buffer during the entire training process. Since ER decay reduces the sampling weight of older data, the sample times for earlier transitions will be lower at first. However, as training continues, the sample times for later transitions will increase. This aligns with the goal of ER Decay: to ensure that transitions collected at different environment steps are sampled similarly, rather than favoring older transitions as in the case of normal ER. We will provide a clearer explanation of the axes in the paper.
> ___
> **Q4:** *"Any intuition for the design of the Network Expansion? Are there any alternative ways to expand the network?"*
>
> **A:** The intuition behind the design of the Network Expansion is based on the concept of infantile amnesia, as mentioned in the introduction. We believe that as data increases, the network's capacity must be expanded to enhance its representation power. In our early experiments, we also found that naive network expansion could improve the agent's sample efficiency in certain environments. There are several alternative ways to expand the network, such as increasing the network's width (e.g., PNN[1], DEN[2]) or using sparse networks and adjusting the network topology (e.g., Neuroplastic Expansion in Deep Reinforcement Learning[3]). However, increasing the depth of the network is the most straightforward and simple approach. When designing our method, we aimed for an easy implementation that could be applied to other algorithms besides OBAC to boost performance, rather than being limited to our specific backbone.
> [1]: Rusu, A. A., Rabinowitz, N. C., Desjardins, G., Soyer, H., Kirkpatrick, J., Kavukcuoglu, K., ... & Hadsell, R. (2016). Progressive neural networks. arXiv preprint arXiv:1606.04671.
> [2]: Yoon, J., Yang, E., Lee, J., & Hwang, S. J. (2017). Lifelong learning with dynamically expandable networks. arXiv preprint arXiv:1708.01547.
> [3]: Liu, J., Obando-Ceron, J., Courville, A., & Pan, L. (2024). Neuroplastic Expansion in Deep Reinforcement Learning. arXiv preprint arXiv:2410.07994.
> ___
> **Q5:** *"line 37, Network Expansion introduces new neurons to the model early in training. Any implementation details about how to decide 'early in training'?"*
>
> **A:** We expand our critic networks at the 50k-th and 250k-th iterations, as detailed in Appendix B.1. These points are considered early in the training process, especially given that the networks are updated up to 2 million times during training.
> ___
> **Q6:** *"What is the final size of the network after training compared to other methods?"*
>
> **A:** The largest network in FoG has a depth of 4, resulting in 9 linear layers, each of size 512 x 512 (calculated as 4 * 2 + 1 = 9). With 9 such networks in OBAC, the total parameter count is approximately 21 million. This is about four times the size of both BRO and SimBa.

---

### Official Review · Reviewer_LRoZ · 2025-03-13

**Overall Recommendation:** 3

**Summary:**

The paper draws inspiration from the phenomenon of infantile amnesia in neuroscience, proposing a "forget and grow" mechanism to mitigate primacy bias in deep reinforcement learning. The authors identify limitations in existing reset mechanisms and introduce two novel strategies: Experience Replay Decay (ER Decay), which gradually reduces the influence of older experiences, and Network Expansion, which introduces new neurons early in training to facilitate adaptation. These methods are integrated into a new algorithm, Forget-and-Grow (FoG). Empirical results across multiple benchmarks demonstrate FoG's superiority over existing methods, establishing it as a competitive approach in sample-efficient deep reinforcement learning.

**Claims And Evidence:**

The empirical results demonstrate that FoG performs well, but the underlying reasons behind the effectiveness of the forget-and-grow mechanism remain insufficiently explored. While I am not requesting a full theoretical analysis, the evidence provided—particularly Figure 10—seems inadequate. A deeper investigation into the effects of network expansion would be valuable for the community. For example, measuring various weight and gradient norms or incorporating plasticity metrics (e.g., dormant neurons) could provide better insights into what happens when new neurons are introduced.

Regarding the forgetting mechanism, my interpretation is that ER Decay shifts learning dynamics toward on-policyness rather than purely off-policy. However, I am curious whether ER Decay is truly the key factor in achieving this effect. A natural baseline to consider would be a smaller replay buffer—does it lead to similar on-policyness effects and performance improvements? While I find the motivation for on-policyness convincing, as a practitioner, I would like to understand whether ER Decay is the best approach or if there are simpler alternatives with comparable benefits.

For the growth mechanism, the comparison with plasticity injection appears somewhat misleading. Plasticity injection was originally designed to diagnose plasticity loss, not to improve performance, so stating that it is "complex" seems unfair. Instead, this paper can be viewed as an extension of plasticity injection, applying it to intermediate layers rather than just a specific subset of the network. I recommend revising the framing of this comparison to reflect the relationship between these methods better. Additionally, after reviewing the appendix, it is unclear whether the authors expanded the network in depth or width—clarifying whether new layers were appended or if existing layers were widened would improve transparency.

**Essential References Not Discussed:**

It would be beneficial to incorporate relevant prior work on progressive neural network architectures in continual learning and reinforcement learning, ensuring a more comprehensive discussion.

- Progressive Neural Networks, Rusu et al, arXiv'16.
- Lifelong learning with dynamically expandable networks, Yoon et al, ICLR'18.
- PLASTIC: Improving Input and Label Plasticity for Sample Efficient Reinforcement Learning, Lee et al, NeurIPS'23.
- Mixtures of Experts Unlock Parameter Scaling for Deep RL, Johan et al, ICML’24.
- Neuroplastic Expansion in Deep Reinforcement Learning, Liu et al, ICLR'25.
- Towards General-Purpose Model-Free Reinforcement Learning, Fujimoto et al, ICLR’25.

**Experimental Designs Or Analyses:**

- The experimental design could be improved by exploring different network growth strategies, including existing methods (e.g., PNN, DEN, NE), and analyzing learning dynamics using plasticity metrics (e.g., dormant neurons, gradient norms).

- Additionally, the current training setup is complex, making it harder for the community to adopt. I suggest two possible improvements:
   - Simplify the experimental framework: Instead of integrating forgetting and growing mechanisms with multiple training protocols, evaluating them within a simpler baseline (e.g., SimBa or TD-MPC2) would better isolate their contributions and improve clarity.
   - Unify and streamline the training setup: The current reset protocol is overly detailed, using different reset lists for specific benchmarks and tasks (e.g., locomotion tasks in HumanoidBench vs. other environments). A more integrated and standardized reset mechanism, along with a consistent OBAC wait period (e.g., 250k iterations) across all experiments, would improve reproducibility and ease of adoption while maintaining strong performance.

**Methods And Evaluation Criteria:**

The evaluation and comparisons in the paper are limited and require further depth.

A crucial missing comparison is with the Plasticity Injection paper. Given its relevance, a direct empirical and conceptual comparison would strengthen the evaluation. Additionally, there are multiple established methods for expanding neural networks, such as Progressive Neural Networks (PNN) [1], Dynamically Expandable Networks (DEN) [2], and Neuroplastic Expansion (NE) [3] in reinforcement learning. The current discussion does not sufficiently situate the proposed network expansion mechanism within this broader landscape. A more comprehensive analysis is necessary to highlight the novelty and advantages of the approach compared to these existing methods.

In particular, the paper should address how FoG's expansion strategy differs from or improves upon these prior works. Does it offer advantages in terms of computational efficiency, stability, or sample efficiency?

Additionally, the computational cost of increasing network depth needs further discussion. Since deeper networks require more sequential computation, they may introduce significant overhead. It is important to compare the computational cost of FoG against BRO, SimBa, and TD-MPC, as these methods are widely used baselines.

**Other Comments Or Suggestions:**

- In Section 3.1, the concept of expansion appears after the preliminary experiments, making it hard to follow. I suggest introducing expansion earlier for better readability.
- The OBAC backbone is not widely adopted in the community, so a more self-contained explanation would help readers unfamiliar with it.
- Figure resolution is too high, causing scrolling lag—reducing DPI would improve readability.
- In Related Work, SimBa should be described accurately—it promotes simplicity bias rather than alleviating it to enable network scaling.


I appreciate the idea and direction, but the paper needs more completeness. Strengthening connections to related work, providing a deeper analysis of growing effects, and offering a more thorough comparison with existing methods would improve its clarity and impact. I am open to increasing my score if these concerns are well addressed.

**Other Strengths And Weaknesses:**

n/a

**Questions For Authors:**

n/a

**Relation To Broader Scientific Literature:**

n/a

**Theoretical Claims:**

I've checked the proof in the Appendix.

---

> ### Author Rebuttal · Authors · 2025-04-01
>
> We thank the reviewer for the time and effort you have dedicated to reviewing our work. We deeply appreciate your careful and thorough review. In the following, we seek to address each of your concerns.
> ___
> **Q1** *"the evidence provided—particularly Figure 10—seems inadequate"*
>
> **A:** We tracked the ratio of activated neurons during training, and FoG does indeed increase the ratio of activated neurons. This indicates that the model effectively utilizes more of its capacity during training.
>
> Training curves available here: https://anonymous.4open.science/r/ICML-dormant-E41C/README.md.
>
> Moreover, we analyze representations via replay buffer samples processed by the critic's last layer for t-SNE visualization, Fixed 2/4 blocks vs. expanded networks. Network expansion shows clearer clustering and better feature separation/structure.
>
> Visualization results available here: https://anonymous.4open.science/r/ICML-t_SNE-D076/README.md.
> ___
> **Q2:** *"A natural baseline to consider would be a smaller replay buffer—does it lead to similar on-policyness effects and performance improvements? "*
>
> **A:** We appreciate the suggestion to compare ER Decay with a smaller replay buffer. We conducted experiments on SAC with only modifications of replay buffer on 3 tasks. The results show that SAC with ER Decay significantly outperforms those with a smaller buffer size ranging from 5e4 to 5e5.
>
> While ER Decay introduces more "on-policy" characteristics, it also retains older transitions, preventing forgetting and improving final performance.
>
> Training curves available here: https://anonymous.4open.science/r/ICML-buffer_size-5FB0/README.md.
> ___
> **Q3:** *"the comparison with plasticity injection appears somewhat misleading"*
>
> **A:** Your concern is valid—we acknowledge potential imprecision. By "complex," we referred to implementation complexity (parameter freezing, residual construction) versus FoG's direct parameter addition without tricks. We will clarify this distinction and refine descriptions in the revised paper.
> ___
> **Q4:** *"The current discussion does not sufficiently situate the proposed network expansion mechanism within this broader landscape. "*
>
> **A:** We have compared our network expansion mechanism with three related works:
>
> 1) **Neuroplastic Expansion in Deep Reinforcement Learning**
>
>   This paper presents a clever and effective approach by utilizing sparse networks and expanding through network topology adjustments. Our method, better suited for high replay ratios and larger networks, is easier to implement. While we don't have their code, we observe that naively expanding network capacity boosts performance in Gym environments, especially HalfCheetah. However, this success doesn't easily extend to more complex environments.
>
> 2) **Progressive Neural Networks & Dynamically Expandable Networks**
>
>    Both methods are designed for multitask learning, expanding network width when new tasks are introduced, whereas our approach increases depth. Since the settings differ, direct performance comparison is not feasible, but we will discuss these methods in the related works section.
> ___
> **Q5:** *"the computational cost of increasing network depth needs further discussion"*
>
> **A:** FoG's computation is about 4× that of BRO. However, our goal is to explore the limits of effective network scaling. On humanoid tasks, FoG outperforms larger models, including SimBa (10-layer, ~42M params), TD-MPC2 (19M params), and BRO (10-layer, ~42M params), despite having at most 21M params and lower computational cost. We will add a discussion on computational cost in the paper.
>
> Training curves available here: https://anonymous.4open.science/r/ICML-full_size-25B6/README.md.
> ___
> **Q6:** *" the current training setup is complex"*
>
> **A:** Both FoG's network structure and OBAC backbone play significant roles in improving performance. We conducted an ablation study to test FoG without OBAC(training curves: https://anonymous.4open.science/r/ICML-structure-B1DF/README.md). While FoG with SAC still gives strong performance, using OBAC backbone is helpful for both performance and parameter scaling.
>
> Regarding the setup, we already use a consistent OBAC wait period across experiments. The plasticity loss differs for each task, and we haven't yet explored adaptive reset or other methods like partial resetting or noise injection. However, we’ve minimized the number of reset lists to simplify the setup.
> ___
> **Q7:** *"It would be beneficial to incorporate relevant prior work" " I suggest introducing expansion earlier for better readability." "The OBAC backbone is not widely adopted in the community"  "Figure resolution is too high" "In Related Work, SimBa should be described accurately"*
>
> **A:** Thank you for the suggestions. We will add a dedicated section on network expansion in related works and make corresponding adjustments throughout the paper.

---

> > ### Comment · Reviewer_LRoZ · 2025-04-02
> >
> > I have updated my score from 2 to 3. The paper's core message is interesting and valid, which motivated the higher score. However, the writing still leaves significant room for improvement. If accepted, I hope the paper is restructured to clarify its contributions better and differentiate itself from related work.
> >
> > On a personal note, I found the "grow" component particularly compelling. I would accept the paper even without the "forget" part. The paper would be stronger if it rigorously evaluated this component across a variety of architectures and algorithms, demonstrating its potential for integration into the RL community.

---

### Official Review · Reviewer_ttxg · 2025-03-13

**Overall Recommendation:** 3

**Summary:**

The paper addresses the challenge of Deep Reinforcement Learning (DRL) in continuous control tasks, where models suffer from primacy bias, overfitting to older memories in the replay buffer. Drawing inspiration from infantile amnesia in humans, the authors propose two modifications: (1) a decaying replay buffer to mitigate overfitting to older memories, and (2) an increase in network size to enhance continuous learning. These modifications are empirically validated across 41 control tasks, demonstrating their effectiveness in improving learning performance.

**Claims And Evidence:**

- The authors claim that standard experience buffers cause models to overfit to older memories, hindering the learning of new information. This is supported by a critic loss heat map, which visualizes the loss in plasticity over time. However, a more comprehensive comparison across all 41 tasks would strengthen this claim. The analysis can be placed in the appendix.

- The claim that increasing the critic network size improves learning is supported by empirical ablation studies. However, the authors do not provide a representation analysis to explain how the expanded network adapts to new memories without interfering with older ones. For instance, does adding new neural blocks increase interference within the network and noise, hence supporting new learning? Such an analysis would offer deeper insights into the mechanism behind this improvement.

**Essential References Not Discussed:**

- Noise-based regularization techniques, such as reinitializing unused parameters with noise (Dohare et al. 2024, Nature), could enhance continual learning in RL models.
- The hypothesis that noise regularization allows networks to explore degenerate solution spaces and escape local minima (Kumar et al. 2024, bioRxiv) is relevant but not discussed.

**Experimental Designs Or Analyses:**

- An inconsistency arises in Figure 3, where the critic loss for Normal SAC is significantly higher than for other models, yet it still achieves decent task performance. This discrepancy warrants further explanation.

- The authors should analyze the learned representations and their evolution under the proposed modifications (decaying replay buffer and network expansion). Empirical results alone are insufficient to fully validate the mechanisms in improving performance.

**Methods And Evaluation Criteria:**

The critic loss heat map effectively illustrates the loss in plasticity as training progresses. However, additional metrics, such as tracking parameter changes across learning steps (e.g., Kumar et al. 2024, bioRxiv 2024.12.12.627755 Supp. Fig. 4), could provide a more robust evaluation of plasticity loss and the efficacy of the proposed strategies.

**Other Comments Or Suggestions:**

- The title's use of "continuous control" may be misleading, as the framework could potentially apply to discrete control tasks as well.
- The term "scaling" in the title might be misinterpreted as referring to scaling laws in machine learning.
- The consistent dips in reward curves in Figure 8 require further explanation.
- Clarify the difference between Expanded SAC (Figure 3) and FoG.

**Other Strengths And Weaknesses:**

- Strengths: The paper presents a novel approach to addressing primacy bias in DRL, with empirical validation across a wide range of tasks.
- Weaknesses: The number of random seeds used for experiments should be explicitly stated in all figure captions, and error bars should be included in bar graphs e.g. Fig. 1. Increasing the number of random seeds to 10, particularly for Figure 7, would improve the robustness of the results.

**Questions For Authors:**

- Can the authors perform a representation analysis to elucidate how block expansion facilitates learning without interfering with older memories?
0 Could the authors discuss the potential benefits of injecting noise into network parameters, as suggested by recent literature?

**Relation To Broader Scientific Literature:**

The paper draws an interesting analogy between infantile amnesia in humans and primacy bias in DRL. However, it does not discuss recent advancements in noise-based regularization techniques, such as those proposed by Dohare et al. (2024, Nature) and Kumar et al. (2024, bioRxiv), which could offer additional insights into continual learning and flexibility in RL models.

**Theoretical Claims:**

The theoretical claims regarding oversampling older memories appear correct and are well-supported.

---

> ### Author Rebuttal · Authors · 2025-04-01
>
> We thank the reviewer for the time and effort you have dedicated to reviewing our work. We deeply appreciate your careful and thorough review. In the following, we seek to address each of your concerns.
> ___
> **Q1:** *"The authors should analyze the learned representations and their evolution under the proposed modifications (decaying replay buffer and network expansion).."*
>
> **A:** We tracked the ratio of activated neurons during training, and FoG does indeed increase the ratio of activated neurons. This indicates that the model effectively utilizes more of its capacity during training.
>
> For the training curve, please refer to the following link: https://anonymous.4open.science/r/ICML-dormant-E41C/README.md.
>
> Due to time constraints, we cannot complete the comparison across all 41 tasks immediately, but we will conduct it in the future and add the results to the appendix.
> ___
> **Q2:** *" The authors do not provide a representation analysis to explain how the expanded network adapts to new memories without interfering with older ones."*
>
> **A:** We analyze representations via replay buffer samples processed by the critic's last layer for t-SNE visualization, Fixed 2/4 blocks vs. expanded networks. Network expansion shows clearer clustering and better feature separation/structure.
>
> For the visualization results, please refer to: https://anonymous.4open.science/r/ICML-t_SNE-D076/README.md.
>
> Due to time constraints, we currently provide results for HalfCheetah-v4 at 200k steps, but will release more within a week.
> ___
> **Q3:** *"An inconsistency arises in Figure 3, where the critic loss for Normal SAC is significantly higher than for other models, yet it still achieves decent task performance."*
>
> **A:** We sincerely apologize for the color mislabeling of SAC and SAC with reset in the left-bottom plot of Figure 3. However, this does not affect the validity of our experimental results.
>
> Even though SAC performs reasonably well, it still performs the worst among all variants, achieving less than half the performance of Expanded-SAC. Its decent early-stage performance stems from the effective use of initial data, but its reward growth stagnates later due to poor utilization of new data.
>
> Additionally, SAC shows low loss for old data but significantly higher loss for new data, supporting our claim that standard experience buffers overfit to early memories, hindering new learning.
> ___
> **Q4:** *"However, it does not discuss recent advancements in noise-based regularization techniques."*
>
> **A:** These techniques are indeed effective in mitigating plasticity loss. However, they are orthogonal to the ER Decay and network expansion proposed in our work, and their combined effects remain an open question for future exploration. While FoG does not incorporate these techniques, we will still discuss them in the related works section to provide a more comprehensive perspective.
> ___
> **Q5:** *"The number of random seeds used for experiments should be explicitly stated in all figure captions, and error bars should be included in bar graphs e.g. Fig. 1."*
>
> **A:** We are currently running 16 tasks for the main result in Figure 7, with each task using 10 seeds. However, due to time constraints, we have not yet obtained the complete results. We will release the results within one week.
>
> Currently we have finished with 4 tasks, please check https://drive.google.com/drive/folders/14u4Ag3MrD9GVl4pib5RneSo6n5WGeQO_ for training curves.
> ___
> **Q6:** *"The title's use of "continuous control" may be misleading"*
>
> **A:** Thank you for the suggestion. However, we have only evaluated FoG on continuous control benchmarks and have not yet explored its performance on discrete control tasks. We primarily followed BRO and SimBa, which also focus on continuous control.
> ___
> **Q7:** *"The term "scaling" in the title might be misinterpreted"*
>
> **A:** Thank you for raising this important point. We acknowledge that the term "scaling" in the title could indeed be conflated with scaling laws in machine learning (ML), and we will revise the title in subsequent versions to use clearer phrasing, such as "Scaling Up Parameters" or "Bigger Models," to explicitly distinguish our focus.
> ___
> **Q8:** *"The consistent dips in reward curves in Figure 8 require further explanation"*
>
> **A:** The dips in Figure 8 occur when the network is reset, requiring it to relearn from the replay buffer, causing a temporary drop before recovery.
> ___
> **Q9:** *"Clarify the difference between Expanded SAC (Figure 3) and FoG."*
>
> **A:** The difference between Expanded SAC and FoG lies primarily in the use of the OBAC backbone and some slight modifications to the network structure in FoG. We will provide a clearer and more detailed explanation of these differences in the paper and will also modify the names of these SAC variants to make them easier to understand.

---

### Official Review · Reviewer_TdFu · 2025-03-14

**Overall Recommendation:** 2

**Summary:**

This paper focuses on the problem of sample efficiency in deep reinforcement learning. The authors introduce the Forget and Grow (FoG) method, which relies on three ideas: 1) reducing the sampling probability of older samples, 2) expanding the network size, and 3) resetting the network with a certain frequency. The results suggest that FoG outperforms multiple performant methods, namely Simba, BroNet, and TDMPC2 in a number of environments.

**Claims And Evidence:**

While some of the claims are supported with evidence, many are problematic. Here I focus on a few:
- The comparison against other methods may be unfair since FoG uses increased computation while the other does not. One approach to improve the empirical evaluation is to use other methods that use the maximum network size used by FoG.
- It’s not clear what defines old and new experience, so it is hard to just make claims about deemphasizing the probability of old data in favor of new data.
- The authors focus on the two mechanisms they propose: experiencing replay decay and growing the network. However, there is a missing mechanism which is reset. Resetting is part of FoG, but it is not emphasized as the other new mechanisms, which might be confusing. An experiment is needed to show how FoG performs without resets.

**Essential References Not Discussed:**

The paper “Neuroplastic Expansion in Deep Reinforcement Learning” by Jiashun Liu, Johan Obando-Ceron, Aaron Courville, and Ling Pan is based on growing the network along with some experience replay techniques.

**Experimental Designs Or Analyses:**

Check the Methods And Evaluation Criteria section.

**Methods And Evaluation Criteria:**

- The paper uses standard benchmarking tasks that are well-accepted to study the problem of sample efficiency in deep RL methods.
- However, the paper misses comparison against similar methods. For example, the paper “Neuroplastic Expansion in Deep Reinforcement Learning” by Jiashun Liu, Johan Obando-Ceron, Aaron Courville, and Ling Pan is based on growing the network along with some experience replay techniques.
- Additionally,  it’s not clear that the evaluation is fair if FoG uses much more computation than other methods.
- Additionally, using the critic loss for comparison in Figure 3 and Figure 5 is problematic since the loss doesn’t necessarily represent any meaningful results. For example, a method that gives zero-value function prediction everywhere would achieve the maximum score based on the metric you’re proposing, which shows that it’s problematic.
- Finally, the experiments are conducted with only three independent runs (seeds), which is a very small number of runs. Many of the figures have overlapping confidence intervals, so their statistical significance may be compromised.

**Other Comments Or Suggestions:**

N/A

**Other Strengths And Weaknesses:**

The paper considers a fundamental problem in deep reinforcement learning. The idea seems exciting but evaluation is lacking rigor. Overall, I would like the paper to be published, but I think it’s not ready in its current format to be published in ICLR 2025, and thus my recommendation is to reject it.

Additionally, the parallel with infantile amnesia needs to be de-emphasized in the paper since it doesn’t match the natural phenomenon. The brain doesn’t only contain neurogenesis but also neural pruning, which breaks the parallel since FoG does not prune its network.

**Questions For Authors:**

- How does FoG perform without resetting?
- How does FoG perform to the Neuroplastic Expansion (NE) method?

**Relation To Broader Scientific Literature:**

The paper considers an important research question about sample-efficient deep RL methods, which is of interest to many people in the research community.

**Theoretical Claims:**

No theoretical claims are presented in the paper apart from the ones used to motivate intuition.

---

> ### Author Rebuttal · Authors · 2025-04-01
>
> We thank the reviewer for the time and effort you have dedicated to reviewing our work. We deeply appreciate your careful and thorough review. In the following, we seek to address each of your concerns.
> ___
> **Q1:** *"The comparison against other methods may be unfair since FoG uses increased computation while the other does not."*
>
> **A:** Our experiments on the humanoid benchmark (h1-walk & h1-run) demonstrate that FoG outperforms competitive baselines even when compared to larger models. We tested SimBa (depth=10, ~42M params), TD-MPC2 (19M version), and BRO (depth=10, ~42M params)—all of which are larger than FoG (at most 21M params). Despite this, FoG achieves superior performance while using less computation in this setup.
>
> Moreover, TD-MPC2 and BRO fail to improve over their default sizes, and only SimBa shows an improvement in h1-walk but not in h1-run, indicating that comparing FoG to these baselines at their default sizes is fair. This further highlights FoG’s ability to efficiently manage a larger number of parameters.
>
> For additional insights, please refer to the training curve figure:https://anonymous.4open.science/r/ICML-full_size-25B6/README.md.
> ___
> **Q2:** *"It’s not clear what defines old and new experience"*
>
> **A:** The distinction between old and new experience is based on the time at which the data was collected and added to the replay buffer. The earlier the data was collected and stored in the buffer, the older it is. ER Decay allows agents to focus on ~100k transitions that's newly collected to buffer. We will provide a clearer explanation in the paper.
> ___
> **Q3:** *" An experiment is needed to show how FoG performs without resets."*
>
> **A:** In the scenario without resets, FoG still outperforms SimBa and BRO with the maximum network size (~42M) on humanoid-run, humanoid-walk, and HalfCheetah-v4, while using less computation. However, removing resets hinders FoG’s performance, showing a clear drop compared to our original version. The reset mechanism was introduced to increase the replay ratio, allowing better utilization of data collected, and hence maximizing data efficiency.
>
> For details, refer to the training curve figure:  https://anonymous.4open.science/r/ICML-no_resets-76A1/README.md.
> ___
> **Q4:** *"the paper misses comparison against similar methods."*
>
> **A:** We currently do not have access to the code from the referenced paper. However, as demonstrated in their work, we also observe that naively expanding network capacity significantly improves performance in many Gym environments, especially HalfCheetah. Yet, this success does not easily extend to more challenging environments like DMC, MetaWorld, or the humanoid benchmark. While increasing the replay ratio enhances sample efficiency, this approach alone is not sufficient to prevent early convergence.
> ___
> **Q5:** *"using the critic loss for comparison in Figure 3 and Figure 5 is problematic"*
>
> **A:** Your point is valid, relying solely on critic loss is insufficient. However, SAC's critic loss in this experiment obviously exceeds reasonable thresholds. Such abnormally large errors can destabilize actor gradients and impede training.
>
> To further substantiate our claim, we tracked the ratio of activated neurons[1]. FoG does indeed increase the ratio of activated neurons. This indicates that the model can effectively utilizes more of its capacity during training, achieving better learning efficiency[2].
>
> For the dormant ratio curves, please refer to the following link: https://anonymous.4open.science/r/ICML-dormant-E41C/README.md.
>
> [1]:Liu, J., Obando-Ceron, J., Courville, A., & Pan, L. (2024). Neuroplastic Expansion in Deep Reinforcement Learning. arXiv preprint arXiv:2410.07994.
> [2]:Xu, G., Zheng, R., Liang, Y., Wang, X., Yuan, Z., Ji, T., ... & Xu, H. (2023). Drm: Mastering visual reinforcement learning through dormant ratio minimization. arXiv preprint arXiv:2310.19668.
> ___
> **Q6:** *"the experiments are conducted with only three independent runs (seeds), which is a very small number of runs. "*
>
> **A:** We are currently running 16 tasks for the main result in Figure 7, with each task using 10 seeds. However, due to time constraints, we have not yet obtained the results. We will release the results within one week.
>
> Currently we have finished with 4 tasks, please check https://drive.google.com/drive/folders/14u4Ag3MrD9GVl4pib5RneSo6n5WGeQO_ for training curves
> ___
> **Q7:** *"the parallel with infantile amnesia needs to be de-emphasized in the paper"*
>
> **A:** Your point is valid. FoG lacks neural pruning, unlike the brain's dual process. However, the analogy with infantile amnesia stemmed from its link to neurogenesis-disrupted memory connections, mirroring network expansion (supported by our neuron activation experiments). We will de-emphasize this parallel in revisions and clarify to prevent misinterpretation.

---

> > ### Comment · Reviewer_TdFu · 2025-04-07
> >
> > I would like to thank the authors for their response. I think some concerns have been addressed, but the majority of them have not been addressed. I will keep my score since 1) There is still no comparison with similar methods. I think it's feasible to implement the baseline method following the pseudocode in the paper even if their code is not available, and 2) Relying on critic loss for evaluation. The authors didn't convince me why it is a viable metric. In fact, the authors agreed. However, they didn't offer an alternative way to fix it such as modifying the experiment or removing it altogether.

---

> > > ### Author Response · Authors · 2025-04-08
> > >
> > > Thank you for your thoughtful feedback. We'd like to address the two main concerns raised:
> > >
> > > 1. **Comparison with Similar Methods**
> > >    We have now included a comparison between FoG and the Neuroplastic Expansion (NE) algorithm using its official implementation. The evaluation spans four tasks—HalfCheetah-v4, dog-run, dog-walk, and humanoid-walk—with both the original NE model and an expanded-capacity variant. Results show that FoG consistently outperforms NE, even at relatively low update-to-data ratio. While increasing NE’s model size brings only marginal improvement. This highlights FoG’s superior scalability and adaptability.
> > >
> > >    For details, please refer to the training curve figure: https://anonymous.4open.science/r/ICML-NE_compare-D403/
> > >
> > > 2. **Use of Critic Loss for Comparison**
> > >    While we do acknowledge that  **critic loss alone** is insufficient to measure plasticity loss, **critic loss is still a very important metric for measuring stability of training and critics' plasticity**. If a low and uniform critic loss curve is not enough to demonstrate plasticity of critics, at least we can be sure that an extremely large critic loss is a definite sign of loss of plasticity in training, as a large loss shows that critic networks can no longer fit transitions in the buffer. That's exactly the case in our original experiments in the paper, SAC's critic loss **obviously exceeds reasonable thresholds**, hence impedes training. While FoG's critic loss is uniform and within a reasonable range.
> > >
> > >     Additionally, as mentioned earlier, we report the ratio of activated neurons of critics in the same tasks to support our viewpoint, following prior work [1, 2]. FoG significantly increases neuron utilization during training, indicating enhanced learning efficiency beyond just loss metrics. As the ratio of activated neurons is a commonly used metric in similar methods, it's widely acknowledged that a lower ratio of dormant neurons indicates better plasticity and learning capacity.
> > >
> > >    For more details, please refer to the dormant ratio curves: https://anonymous.4open.science/r/ICML-dormant-E41C/README.md.
> > >
> > >     **We strongly believe that critic loss accompanied by the ratio of activated neurons can effectively support our claim** and we are happy to answer any concerns or doubts of yours.
> > >
> > > **Thank you for raising your score! If you have any more concerns or doubts, we are very willing to discuss!**
> > >
> > >    [1] Liu et al. (2024). Neuroplastic Expansion in Deep Reinforcement Learning.
> > >    [2] Xu et al. (2023). Dormant Ratio Minimization for Efficient Representation Learning.

---

### Decision · Program_Chairs · 2025-05-01

**Decision:**

Accept (poster)

**Comment:**

This paper presents an algorithm attacking the issue of primacy bias in RL inspired by neuroscience. The proposed techniques appear to be sound, but it is unclear exactly what the neuronal mechanism at play here is. Experiments show improvements over baselines, but with fewer random seeds than we would like. Additional desired baselines were provided during the rebuttal period. Therefore, I would recommend weak accept.

I would also encourage the authors to include an impact statement in their next draft.